**Data Availability Statement:** The data used in the study are openly available to the public without application or screening for access come from the

# An integrated risk and epidemiological model to estimate risk-stratified COVID-19 outcomes for Los Angeles County: March 1, 2020—March 1, 2021

**Abigail L. Horn**[1]*, **Lai Jiang**[1], **Faith Washburn**[2], **Emil Hvitfeldt**[1], **Kayla de la Haye**[1], **William Nicholas**[2], **Paul Simon**[2], **Maryann Pentz**[1], **Wendy Cozen**[1], **Neeraj Sood**[3,4], **David V. Conti**[1]

1 Department of Preventive Medicine, Keck School of Medicine, University of Southern California, Los Angeles, CA, United States of America, 2 Los Angeles County Department of Public Health, Los Angeles, CA, United States of America, 3 Sol Price School of Public Policy, University of Southern California, Los Angeles, CA, United States of America, 4 Schaeffer Center for Health Policy and Economics, University of Southern California, Los Angeles, CA, United States of America

* abigail.horn@usc.edu

## Abstract

The objective of this study was to use available data on the prevalence of COVID-19 risk factors in subpopulations and epidemic dynamics at the population level to estimate probabilities of severe illness and the case and infection fatality rates (CFR and IFR) stratified across subgroups representing all combinations of the risk factors age, comorbidities, obesity, and smoking status. We focus on the first year of the epidemic in Los Angeles County (LAC) (March 1, 2020–March 1, 2021), spanning three epidemic waves. A relative risk modeling approach was developed to estimate conditional effects from available marginal data. A dynamic stochastic epidemic model was developed to produce time-varying population estimates of epidemic parameters including the transmission and infection observation rate. The epidemic and risk models were integrated to produce estimates of subpopulation-stratified probabilities of disease progression and CFR and IFR for LAC. The probabilities of disease progression and CFR and IFR were found to vary as extensively between age groups as within age categories combined with the presence of absence of other risk factors, suggesting that it is inappropriate to summarize epidemiological parameters for age categories alone, let alone the entire population. The fine-grained subpopulation-stratified estimates of COVID-19 outcomes produced in this study are useful in understanding disparities in the effect of the epidemic on different groups in LAC, and can inform analyses of targeted subpopulation-level policy interventions.

## 1 Introduction

Health disparities have emerged with the COVID-19 epidemic because the risk of exposure to infection and the prevalence of risk factors for severe outcomes given infection vary within

GitHub page of the Los Angeles Times Data and Graphics Department. In addition, these data are provided in a public repository at: https://github.com/AbigailHorn/COV2-LA/tree/master/data.

**Funding:** ALH was supported from an NIH Ruth L Kirschstein National 591 Research Service Award (NRSA) Institutional Training Grant T32 5T32CA009492-35. 592 DVC was supported by NIH P01CA196569 and NCI P30CA014089. EH was supported 593 by NIH P01CA196569. DVC and ALH are also funded by COVID-19 Keck Research 594 Fund from the University of Southern California.

**Competing interests:** The authors have declared that no competing interests exist.

and between populations and over time [1–7]. For public health policy makers to better address the pandemic, models reporting stratified estimates are necessary to investigate the potential outcomes of policy scenarios targeting specific subpopulations. However, estimated epidemic quantities such as rates of severe illness and death, the case fatality rate (*CFR*), and the infection fatality rate (*IFR*) are often expressed in terms of aggregated population-level estimates or by age group alone due to the lack of epidemiological data at the refined subpopulation level [8–10]. While data may be available for single risk factor strata such as by age [11], data on subpopulations representing individuals with combinations of risk factors are not reported or available. Conventionally, estimates of risk effects and outcomes given combinations of conditions are obtained through access to individual-level data and the application of multiple regression techniques [5, 12]. At the time of this study, individual-level COVID-19 data were not widely available nor sampled in an appropriate manner to avoid substantial bias [13].

In this paper we develop a model that produces stratified estimates of the probability of disease progression and death for subpopulations representing individuals with combinations of risk factors important for COVID-19 using dynamic epidemiological data at the aggregated population level [14], published studies on the risk of individual risk factors on illness severity, and prevalences of risk factors in the general population. In the absence of access to individual-level data, we apply a statistical technique developed for joint analysis of marginal summary statistics (JAM) [15] to obtain estimates of the conditional effects of combinations of COVID-19 risk factors on the probability of developing severe illness and death, using data from published studies reporting the marginal effects of individual risk factors [2, 3]. We consider the risk factors age, existing comorbidities, obesity, and smoking. Separately, we develop a stochastic epidemic model and use Bayesian methods to estimate time-varying probabilities of hospitalization, ICU admission, and death given infection at the population level. We integrate the conditional risk effects and the population-level probabilities, together with available dynamic data on the prevalence of infections and deaths stratified by age, to estimate the probability of disease progression, *CFRs* and *IFRs*, stratified across all plausible combinations of the modeled risk factors. This approach allows us to produce risk-stratified estimates without access to either individual-level data on disease progression, or subpopulation-level dynamics of infections, hospitalizations, ICU admissions, and deaths by risk groups.

Focusing on Los Angeles County (LAC), the most populous and one of the most diverse counties in the United States, we analyze the estimated overall and risk-stratified time-varying disease progression probabilities, *CFRs*, and *IFRs* in relation to the epidemic timecourse and implemented policy decisions through the first year of the epidemic, from March 1, 2020 through March 1, 2021. Our analysis is framed in terms of three epidemic waves experienced in LAC; a first wave from March 1—May 6, 2020, diminished through a strict lockdown; a second and larger wave, May 7—October 14, 2020, which peaked at the end of July; and a third significantly large wave that began on October 15, 2020, peaked in mid-January, and had subsided by March 1, 2021.

The integrated model allows the comparison of dynamic outcomes and parameters across the overall population, age groups, and more fine-grained subpopulations in LAC representing age and combinations of other risk factors for severe COVID-19 illness. Such fine-grained results can be useful in understanding disparities in the effect of the epidemic on different groups in LAC and can inform studies involving targeted subpopulation-level policy interventions [16].

## 2 Methods

We developed a single-population stochastic dynamic epidemic model that accounts for observed and unobserved transmission of COVID-19 and trajectories through the healthcare system with hospitalization, ICU admission, and death. Using Bayesian methods for parameter estimation and uncertainty quantification, we estimated the population-average time-varying probabilities of transitions between the infected, hospitalized, ICU, death, and recovery compartments, and the resulting population-average time-varying case fatality rate (*CFR*, defined as deaths over observed infections) and infection fatality rate (*IFR*, defined as deaths over all infections) (Section 2.1). In parallel, we used available data from published studies on the *marginal* effects of individual risk factors (age, existing comorbidities, obesity, smoking) to calculate *conditional* risk effects estimates for three models: (1) hospitalization given infection, (2) ICU admission given hospitalization, and (3) death given ICU admission. The conditional risk estimates were integrated with the corresponding probability estimates $\hat{\alpha}_t$, $\hat{\kappa}_t$, and $\hat{\delta}_t$ from the dynamic epidemic model to create a *risk model* (Section 2.2). The *risk model* enables us to estimate, stratified across 54 combinations of the levels of the modeled risk factors (i.e. *risk profiles*), the probability of each stage of disease given infection within LAC. Finally, we integrate the time-varying stratified probability of each stage of disease with the timeseries of observed infections, estimated total infections including observed and unobserved, deaths, together with available data on the prevalence of infections stratified by age, to estimate the risk profile-stratified CFR and IFR across time (Section 2.2.5).

### 2.1 Epidemic model

We develop a model of COVID-19 transmission in a single, homogeneously-mixed population divided into nine compartments representing different disease states (Fig 1). Compartments relating to the transmission of infection are the widely-used susceptible (*S*), exposed (latent but not yet infectious) (*E*), infectious and observed (*I*), and recovered (*R*) classes. By including the exposed compartment, we are able to model the delay between individuals being exposed to infection and becoming infectious. We also include a compartment representing infectious individuals with unobserved and/or unconfirmed infections (*A*). *I* represents cases of infection that have tested positive for the SARS-CoV2 virus and are confirmed in the official register of infection case data. *A* represents cases that are symptomatic but do not appear in the confirmed case data, whether because they are asymptomatic, are symptomatic and do not get tested, or get tested and have a false negative result. We model healthcare utilization and outcome at a more granular level by including compartments representing individuals that are in hospital (*H*), in ICU care (*Q*), and that die (*D*). *H* includes individuals that are receiving care services in skilled nursing facilities (i.e., SNFs). *D* represents only deaths that are confirmed as being COVID-19 related. Each individual can only be in one state at each point in time.

Our model applies the following logic and assumptions. Susceptible individuals will become exposed *and* develop infection (emphasizing that exposure to the virus is not a sufficient condition for developing an infection) and move to the exposed but latent state *E*, meaning they will become, but are not yet, infectious. The transfer of susceptible individuals into the exposed state happens at a *per capita* rate $\beta_t$, the *transmission rate*, defined as the average number of individuals that an infected individual will infect per day. $\beta_t$ controls the rate of disease spread and reduces following modifications including non-pharmaceutical interventions (NPIs). By including the exposed compartment, we are able to model the delay between individuals being exposed to infection and becoming infectious. From the exposed and latent state, individuals will transition to one of the two active infection states: a time-varying fraction $r_t$ of these cases will transfer to the observed infectious state *I*, and the remaining $1 - r_t$ will transfer to the

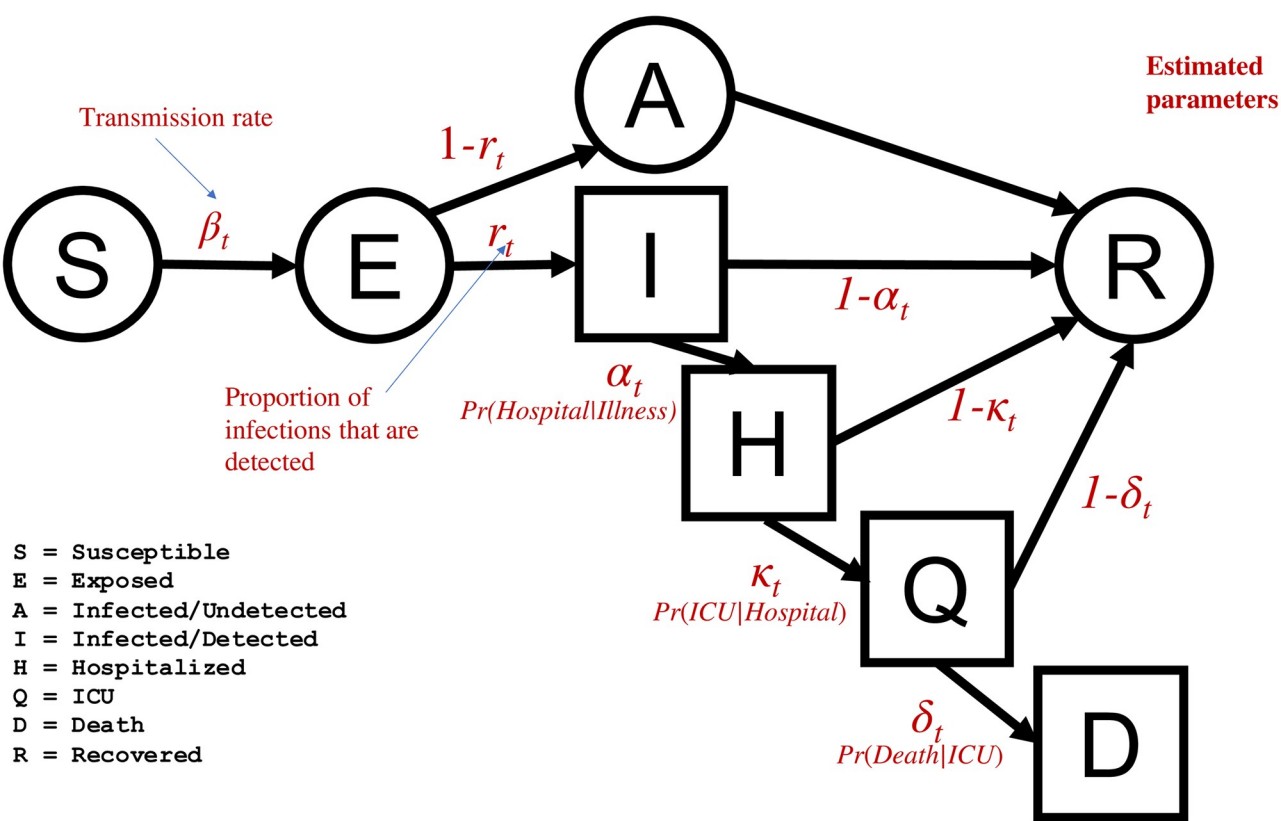

**Fig 1. Epidemic model structure and estimated parameters.** Model compartments with available data are represented as square compartments.

unobserved infectious state $A$. Therefore, the parameter $r$ represents the fraction of all infectious cases that are observed and confirmed. We assume that new infections are created only by individuals in the infected classes ($I$ and $A$), and that individuals in all other compartments, including in hospital, do not contribute to transmission. Individuals transfer from the exposed to infectious and observed ($I$) or unobserved ($A$) compartments at a rate equal to the inverse of the mean latency period, $d_{EI}$.

Infectious cases may either move directly into the recovered state ($R$), or into hospitalization ($H$) if further care is required. Of all observed and infectious cases ($I$), we assume that individuals will require hospitalization with probability $\alpha_t$, equal to $P_t(H|I)$. Infectious individuals transition into the hospitalized state at a rate equal to the inverse of the time between infectiousness and hospitalization, $d_{IH}$, or move directly to recovery at a rate equal to the inverse of the mean time of infection given that hospitalization is not required, $d_{IR}$. Hospitalized individuals will require ICU care with a probability $\kappa_t$, equal to $P_t(Q|H)$, and transfer into $Q$ at a rate equal to the inverse of the mean time in hospital given that ICU care will be required, $d_{HQ}$. With probability $1 - \kappa_t$ they will recover and move into $R$ at a rate of the inverse of the mean time in hospital given that ICU care will not be required before recovery, $d_{HR}$. Individuals in the ICU will recover with probability $1 - \delta_t$, moving to $R$ at rate equal to the inverse of the mean time in ICU before recovery, $d_{QR}$, or will die with probability $\delta_t$, equal to $P_t(D|Q)$, moving to $D$ with rate equal to the inverse of the mean time in ICU care given a fatal case, $d_{QD}$. We assume that all unobserved infections ($A$) will recover directly, since admission to hospital would entail a COVID-19 test. These cases transition to $R$ at the same rate as observed infectious ($I$) individuals, $1/d_{IR}$. We assume recovered individuals cannot be

reinfected due to immunity, and cannot infect others. While the dynamics of transitions within the healthcare setting would change if hospital and ICU, capacity are reached, we do not model this condition. The only route to death is through an observed infection followed by hospitalization and ICU care, meaning we do not model individuals that die from COVID-19 illness at home rather than at a point-of-care. We justify this assumption because the majority of confirmed COVID-19 deaths cases result from individuals who die in SNF, hospital, or following a stay in hospital; analysis of death certificate data from California indicates that 4%-9% of official COVID-19 deaths have occurred at home, across the three epidemic waves (E. Garcia, personal communication based on unpublished data for the state of California, April 20, 2021). Furthermore, we do not model a route to death for individuals without a confirmed COVID-19 infection (*A*), since record of confirmed COVID-19 infection (or probable based on clinical evidence) is needed to be classified as COVID-19 mortality [17].

To model this dynamical state system we employ a discrete-time approximation to the corresponding stochastic continuous-time Markov process in which transitions of individuals between disease stages are seen as stochastic movements between the corresponding population compartments with random transition rates [18, 19]. This model keeps track of the number of individuals in each compartment and the flows of individuals transitioning between compartments through a set of coupled discrete-time multinomial counting processes with transmission rates defined by Poisson processes. To simulate from this system we employ a Euler numerical scheme for Markov process models [18]. For more details see Section 1.5 in S1 Appendix.

The reproductive number, $R_t$, defined as the mean number of secondary cases generated by a typical infectious individual on each day in a fully susceptible population [20], is a function of model parameters including the transmission rate $\beta_t$ [21], and like $\beta_t$, changes in time with behavior and interventions. We use the Next Generation Matrix approach to solve for the reproductive number (Section 1.6 in S1 Appendix) and find that,

$$R_t = \beta_t \left[ \frac{r_t}{\frac{\alpha_t}{d_{IH}} + \frac{1-\alpha_t}{d_{IR}}} + (1 - r_t)d_{IR} \right]. \qquad (1)$$

Thus, $R_t$ is a function of the transmission rate $\beta_t$ as well as other model parameters.

As the pandemic continues and the susceptible population decreases, it becomes important to study the effective reproductive number $R_{eff}$, equal to $R_t$ multiplied by the fraction of the population that is susceptible at time $t$, $S(t)/P(t)$. When $R_{eff} < 1$, the epidemic will begin to decrease (although stuttering chains of epidemic growth may still occur in a stochastic model).

**2.2.1 Parameter estimation.**   All transition rate parameters (e.g., the inverse of the time between exposure and infectiousness $d_E I$) are modeled as fixed values taken directly from published literature (Table 2 in S1 Appendix). The model has five unknown parameters, $\theta = \{\beta_t, r_t, \alpha_t, \kappa_t, \delta_t\}$, which we estimate from COVID-19 data for LAC (Table 3 in S1 Appendix).

Due to the relationships between the five interacting model parameters $\{\beta_t, r_t, \alpha_t, \kappa_t, \delta_t\}$ in the model formulation, a tractable likelihood function was not possible for our model and a likelihood-free method of parameter estimation was required. Furthermore, the formulation of the model allows for multiple parameter solutions to exist. This means that estimated posterior distributions will be multimodal if allowed to vary over a wide prior parameter space. We use a two-step likelihood-free sampling process to define unimodal posterior distributions and achieve convergence in parameter estimates, using a broad grid search followed by approximate Bayesian computation (ABC) sampling. We first perform a broad grid search to identify possible regions for each parameter, from which we decide on a single mode. External data sources were used to specify the parameter range for the grid search (discussed below). Second,

we use ABC sampling to estimate the final posterior distribution for each parameter with a prior distribution informed by the chosen mode from the grid search step. Specifically, we define a prior distribution for ABC as a normal distribution with 95% of its values lying within ±25% of the mean value of the chosen mode; for example, if the mean of a chosen mode for parameter $X$ is determined to be 0.1, then the prior distribution for $X$ will be a normal distribution with standard deviation of 0.01, chosen such that $Pr(0.075 < X < 0.125) \approx 95\%$.

Because prior information from existing studies on the reproductive number $R_t$ is more readily available than for $\beta_t$, we estimate $R_t$ from data and then use Eq 1 to obtain $\beta_t$ from $R_t$. We first estimate $R0$, the initial value of the reproductive number before any interventions. The grid search parameter space for $R0$ is informed by values estimated from previous published studies on COVID-19 [22, 23]. We use geolocation trace data from smartphones, i.e. mobility data, to inform the magnitude and the timing of changes in the distribution of $R_t$ over time from the initial $R0$ value. We incorporate data for LAC provided by Unacast [24] on reductions in distances travelled and encounter rates [25]. Interestingly, this data source diverges from observed trend in infections with the third epidemic wave, demonstrating a decrease in mobility activity as the epidemic surge took off. Thus, for dates after October 15, 2020, we do not use mobility data to inform the grid search space for $R_t$ and instead set this equal to $1 < R_t < R0$, for dates corresponding to increasing infection trends. The grid search parameter space for the fraction of observed cases out of all infections, $r_t$, was informed by results of a CDC study reporting seroprevalence surveys across 10 communities in March through early May 2020 for $t$ within that time period, and was allowed to vary more widely for dates after May 2020 [26]. Grid search ranges for the parameters representing the probabilities of disease stage progression, $\alpha_t$, $\kappa_t$, and $\delta_t$, were informed by the ratios of the observed numbers of infections, hospitalizations, and deaths in LAC. Prior distributions used in parameter estimation are specified in Section 2.1 in S1 Appendix.

The model was fit to the daily and cumulative count of observed infections and deaths, and current numbers in-hospital and in-ICU, coming from the GitHub page of the Los Angeles Times (LA Times) Data and Graphics Department [14]. The infection and death data is sourced from reports logged by LA Times reporters and editors based on reports from the LAC Department of Public Health. The in-hospital and in-ICU data was sourced by the LA Times directly from the California Department of Public Health's Open Data Portal [27]. We use the total of both confirmed and suspected COVID-19 patients in hospital or ICU.

Using ABC on multiple parameters simultaneously produces a joint posterior distribution over all parameters. We simulate the model with each set of jointly-estimated values to produce estimated timeseries of all state variables, as well as to estimate the time-varying case fatality rate, $CFR_t$, and infection fatality rate, $IFR_t$ at each model run. These are calculated as estimated deaths ($D$) over estimated cumulative observed infections ($I$) or estimated cumulative total infections ($I+A$), respectively, on date $t$. Specifically, we simulate the model over 100 jointly estimated parameter sets and 20 stochastic realizations for each set, resulting in 2000 total realizations. We pool together all simulated model trajectories and report their median and 95% credible intervals (CI) as the 2.5th/97.5th quantiles of realizations. This procedure quantifies uncertainty from two sources: variability due to posterior parameter distributions, and variability due to the stochastic variation between model runs with the same parameter values.

## 2.2 Risk model

The *risk model* produces estimates of the probability of disease progression (infection to hospitalization, ICU admission, and death), and the CFR and IFR, stratified across 54 *risk*

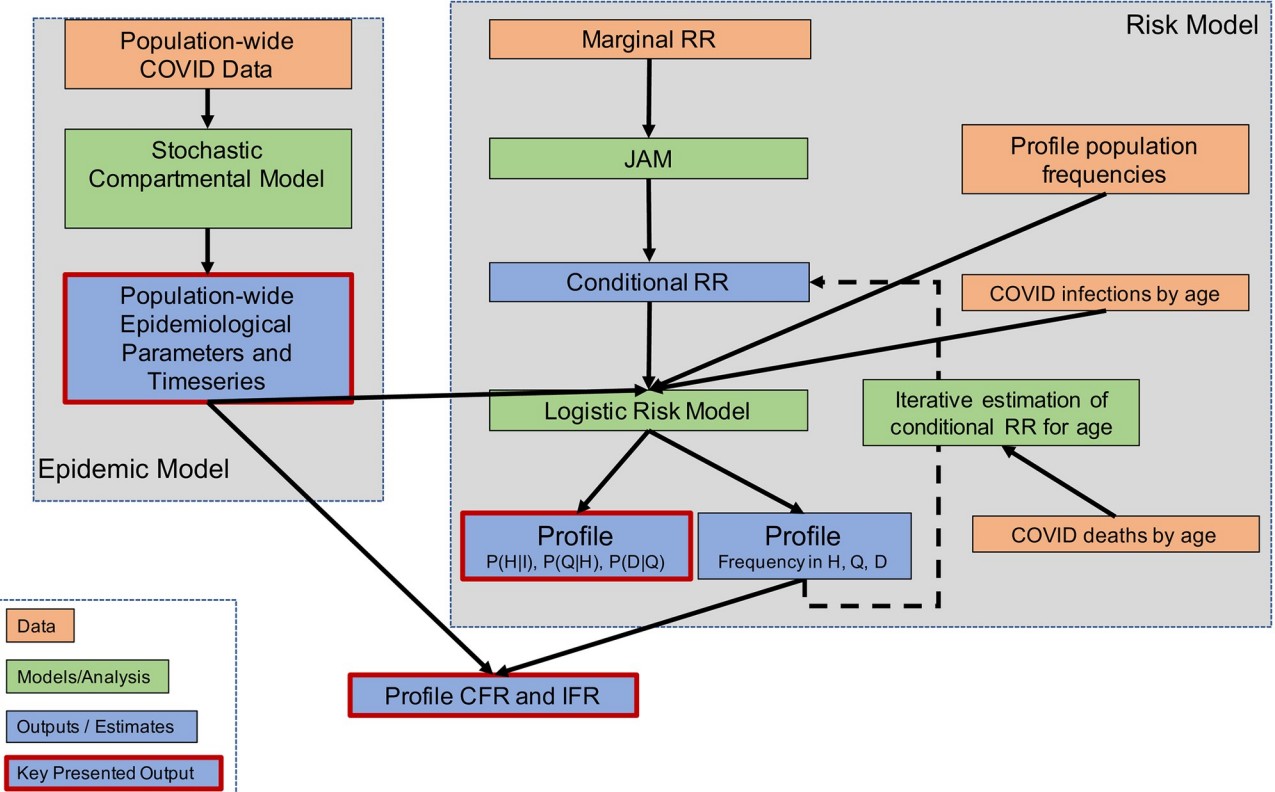

**Fig 2. Flow diagram illustrating the *risk model*, i.e. the set of steps used to produce estimates of *risk profile*-stratified probabilities of disease progression and CFR and IFR.** The diagram shows data inputs, modeling or analysis steps, and outputs or estimates.

*profiles* $q \in Q$, representing all combinations of the different levels of the risk factors age, comorbidities, obesity, and smoking status. The profile-stratified estimates are tied to the LAC population through multiple inputs: they are mean-centered on the overall population epidemic model estimated parameters $\alpha_t$, $\kappa_t$, and $\delta_t$, take into account the prevalence of each risk factor in the LAC population, and are designed to match the prevalence of each *risk profile* over infections and deaths in LAC. The profile-stratified probabilities of disease progression, CFR, and IFR are estimated through six steps, described below and summarized in terms of data inputs, modeling or analysis steps, and outputs or estimates in the flow diagram in Fig 2.

1. Estimate the population-average probability that individuals in LAC who acquire infection are admitted to hospital, $\hat{\alpha}_t$, who are in hospital require admittance to the ICU, $\hat{\kappa}_t$, and who are in ICU will die, $\hat{\delta}_t$, using the epidemic model and Approximate Bayesian Computation as described in the previous section. We also use the epidemic model to estimate the time-series of the numbers of the LAC population that are infected and observed, infected and unobserved, in hospital and ICU, and deaths.

2. Calculate conditional relative risk (RR) estimates for three risk factors (existing comorbidities, obesity, smoking) conditional on one another and on age for three models ($m$): hospitalization given illness, ($H|I$), ICU admission given hospitalization, ($Q|H$), and death given hospitalization, ($D|Q$). This is done using available data from published studies on the marginal RR of the four factors on COVID-19 illness at each stage of disease, the correlation

between these factors in a population resembling that of LAC, and a statistical method called the Joint Analysis of Marginal summary statistics (JAM).

3. Estimate the prevalence of each risk profile $q \in Q$ in the infected population over time. Data inputs to this step are the prevalence of each marginal risk factor in the overall LAC population, the correlation between the risk factors in a population resembling that of LAC, and the proportion of each marginal age group out of all COVID-19 infections in LAC.

4. Estimate the probability of disease stage progression across all risk profiles over time, $P_t\widehat{(H|I)}$, $P_t\widehat{(Q|H)}$, and $P_t\widehat{(D|Q)}$ such that they are mean centered on the probability estimates from the epidemic model $\hat{\alpha}_t$, $\hat{\kappa}_t$, and $\hat{\delta}_t$, respectively. These estimates are produced by integrating $\hat{\alpha}_t$, $\hat{\kappa}_t$, and $\hat{\delta}_t$ (Step 1) with the corresponding conditional risk estimates for each model ($H|I$), ($Q|H$), and ($D|Q$) (Step 2), and the prevalence of each risk profile in the infected population (Step 3) within a logistic model. This step also produces estimates of the frequency of each profile $q \in Q$ at each stage of disease. This is similar in spirit to epidemiologic approaches that combine risk estimates with baseline hazard rates from external sources to estimate absolute risk [28].

5. Estimate the conditional RR of age on the other three risk factors, by iteratively adjusting the conditional RR for each age group input to the logistic risk model (Step 4) until the frequency of each age group over deaths (an output from Step 4) matches the observed distribution of each age age-over-deaths distribution for LAC.

6. Estimate the $CFR_{q,t}$ and $IFR_{q,t}$ for each risk profile using the profile-stratified frequency in infections (Step 3) and deaths (Step 4), and the timeseries of observed infections, unobserved infections, and deaths for LAC overall (Step 1). This is produced by projecting the frequency of each risk profile at each stage of disease (Step 4) onto estimated timeseries of $I$, $I + A$, and $D$, and dividing to obtain the $CFR_{q,t}$ and $IFR_{q,t}$ ratios.

Below, we provide an overview of the methodology used in each step besides Step 1, which was described in Section 2.1. Further details on the methodologies employed in each step are provided in S1 Appendix, Part II. A summary of the mathematical notation used in the risk model is provided on Table 8 in S1 Appendix.

**2.2.1 Step 2: Conditional RR for BMI, smoking, and comorbidities.** The risk factors $p \in P$ included in our analysis are age, body mass index (*BMI*), smoking status (*smoking*), and any comorbidity (*comorbidity*). The comorbidities included are diabetes, hypertension, chronic obstructive pulmonary disease (COPD), hepatitis B, coronary heart disease, stroke, cancer and chronic kidney disease. We modeled age and BMI as an ordinal variable and assume an additive effect of both age and BMI on the three outcomes. Age was categorized within four groups: $0 - 18$, $19 - 49$, $50 - 64$, $65 - 79$, and 80+. BMI was categorized in three groups according to obesity classes: Class 1 (no obesity) $BMI < 30 \frac{kg}{m^2}$; Class 2 (obesity), $30 \leq BMI \leq 40 \frac{kg}{m^2}$; Class 3 (severe obesity), $BMI > 40 \frac{kg}{m^2}$. Any comorbidity and smoking status were modeled as binary variables.

We estimate the conditional RR for BMI, smoking, and comorbidity, conditional on age, for the three models ($H|I$), ($Q|H$), and ($D|Q$) using marginal RR estimates available from reported studies and a method called the Joint Analysis of Marginal Summary Statistics (JAM) [15]. JAM uses two pieces of information: (i) the marginal RR between risk factors and the outcome and (ii) a reference correlation structure between the risk factors, $\Sigma$. For information informing (i) we obtain the marginal log RR between individual risk factors and COVID-19 illness severity from peer-reviewed published COVID-19 studies [2, 3] (left column of Table 1).

**Table 1. The marginal relative risk of each stage of disease collected from published studies on COVID-19 and conditional relative risk estimated by the risk model for each risk factor on rates of hospitalization given infection, ($H|I$); ICU admission given hospitalization, ($Q|H$); and death given ICU admission, ($D|Q$) (95% credible interval).** The reference group is individuals with no comorbidity, $BMI < 30 \frac{kg}{m^2}$, and non-smoking.

| Risk Factors | Marginal RR (95% CI) | Conditional RR (95% CI) |
|---|---|---|
| ($H|I$) | | |
| Ordinal BMI | 2.98 (2.61, 3.39) | 1.82 (1.06, 3.15) |
| Smoker | 1.40 (0.90, 2.17) | 1.76 (0.21, 14.52) |
| Any comorbidity | 3.18 (2.42, 4.18) | 1.50 (0.59, 3.84) |
| ($Q|H$) | | |
| Ordinal BMI | 1.01 (0.86, 1.18) | 1.05 (0.65, 1.69) |
| Smoker | 1.71 (0.87, 3.38) | 1.61 (1.45, 1.79) |
| Any comorbidity | 1.34 (0.87, 2.06) | 1.02 (0.86, 1.20) |
| ($D|Q$) | | |
| Ordinal BMI | 1[†] | 1.12 (0.73, 1.71) |
| Smoker | 1[†] | 1.96 (1.33, 2.89) |
| Any comorbidity | 1.64 (0.81, 3.32) | 1.05 (0.78, 1.43) |

[†]We set the marginal RR for ordinal BMI and smoker to 1 because we did not find the association between obesity class, smoking status, and the likelihood of death given ICU admission $D|Q$ due to COVID-19 in the published literature.

For (ii), we estimate the correlation structure $\Sigma$ using individual-level data from the National Health and Nutrition Examination Survey (NHANES) from 2017–2018 [29], weighted by race/ethnicity proportions to create a population resembling that of LAC (Section 6 in S1 Appendix).

Using the marginal summary statistics from (i), specifically the marginal log relative risks $\psi_{p,m}^{Marg}$ for risk factor $p$ and model $m$, JAM obtains conditional log relative risks $\psi_{p,m}^{Cond}$ for each factor. To accomplish this JAM first expresses the relationship between an outcome $m$, such as hospitalization given infection, ICU admission given hospitalization, and death given ICU admission, and the risk factors $p \in P$ as a normal linear model, $m \sim N(\mathbf{P}\psi, \tau^2\mathbf{I})$. For such a model the conditional or adjusted estimates of effect are given by $\hat{\psi^{Cond}} = (\mathbf{P'P})^{-1}\mathbf{P'm}$. To fit this model without access to individual-level data we substitute an estimate of $\mathbf{P'P}$ based on an estimate of this matrix using the correlation $\Sigma$ between the risk factors from external NHANES data as specified in (ii). $\mathbf{P'm}$ defines the mean value of the outcome for each of the corresponding values of the risk factor. These can be constructed using the marginal log relative risks $\psi_{p,m}^{Marg}$ and the frequencies of each risk factor in the population. (See Section 5 in S1 Appendix for more details).

**2.2.2 Step 3: Prevalence of each risk profile in the infected population.** We estimate the time-varying frequency of each risk profile in the infected population, $\mathbf{f}_{t,q,I}$. First, we estimate the frequency of the risk profiles $q$ in the overall LAC population, $l_q$, by simulating a sample population based on the prevalence of each individual risk factor in LAC, and the weighted correlation structure between the risk factors $\Sigma$ obtained from NHANES data from Step 2. The prevalence of each age group comes from the American Community Survey via the tidy-census R package [30]. The prevalence of obesity, smoking and all comorbidities besides cancer come from the Los Angeles County Health Survey (LACHS), study year 2018 [31]. The prevalence of cancer comes from the California Health Information Survey (CHIS) [32]. Using the vector of prevalences of each risk factor, $\mathbf{l}_p$, and correlation structure $\Sigma$, we generate

a simulated population $\chi$ by sampling from a multivariate normal, $\chi \sim N(x; \mathbf{l}_p, \Sigma)$, where $x$ is the number of samples. An *any comorbidity* variable is constructed as an indicator if any of the comorbidities are present. We then calculate the vector of the frequencies of each risk profile in the overall LAC population, $\mathbf{l}_q$, as its relative frequency in the simulated population $\chi$.

Second, we use COVID-19 infection cases by age group in illnesses [14] together with the estimate of the prevalence of each risk factor in the overall LAC population, $\mathbf{l}_q$, to estimate the frequency of each profile within the infected population on each date. We anchor our estimate of the frequency of each profile over infections on the frequency over ages, as age is the only risk factor with observed infection prevalence data in LAC.

Infection timeseries data by age group come from the LA Times. We use the age group infection numbers from the state of California because data for LAC is not available. We checked that the distribution for aggregate age groups in this California data resemble that reported by the LAC Department of Public Health [33]. To estimate the frequency of each risk profile $q$ in the infected population, $f_{t,q,I}$, the frequency of each age group over infections is stratified across the risk profiles according to the relative frequency of each profile in the baseline LAC population.

**2.2.3 Step 4: Risk-profile-stratified probabilities of disease stage progression.** We construct a logistic model to estimate the probability of disease stage progression across all risk profiles over time, $\widehat{P_t(H|I)}$, $\widehat{P_t(Q|H)}$, and $\widehat{P_t(D|Q)}$. Specifically, the model combines the 54 risk profiles as linear combinations of the risk factors specified in a mean centered design matrix, $\mathbf{X}$; and their corresponding conditional log-RR obtained from JAM, $\hat{\psi}$; with specified intercepts set to the estimated probabilities from the epidemic model (Section 2.1) for $\hat{\alpha}_t$, $\hat{\kappa}_t$, $\hat{\delta}_t$, respectively. For example, to estimate the vector of probabilities of hospitalization given infection for all risk profiles we use $\widehat{P_t(H|I)} = expit(\hat{\alpha}_t + \mathbf{X}\hat{\psi})$. The reference profile are individuals age $0 - 18$ with no comorbidity, $BMI < 30 \frac{kg}{m^2}$, and non-smoking.

The mean-centered design matrix is based on the frequency of each risk profile at each stage of disease. The frequency of each profile in infections comes from Step 3. The frequency at subsequent stages is calculated recursively for each stage of disease (in hospital, in ICU, and deceased) using the frequencies in the previous stage of disease and the calculated incoming probabilities.

**2.2.4 Step 5: Conditional RR for age.** Rather than estimate the conditional RR for age using the same methodology as for the other factors as described in Step 2, we estimate the conditional RR of each age group separately since we have observed data on the distribution of each age group over deaths for LAC. Given this information, we aim to find the solution set that minimizes the distance between the distribution over deaths produced by the logistic model and the observed distribution. Specifically, we choose the conditional RR for age such that the distance between the frequency of each age group over deaths produced in Step 4 and the observed distribution of each age group over deaths in LAC is minimized. This is done through an iterative optimization process in which we vary over a wide search space the conditional RR for each age group for each model that are input to the logistic model (Step 4) and find the maximizing values for the conditional RR.

**2.2.5 Step 6: Risk-profile-stratified $CFR_{q,t}$ and $IFR_{q,t}$.** To calculate the time-varying $CFR_{q,t}$ and $IFR_{q,t}$ for each risk profile, the estimated frequency of each profile in the infected population and in the deceased population (obtained from Step 4) are multiplied by each value of the estimated cumulative number of observed infections ($I$) or total infections ($I+A$), and deaths ($D$) obtained from each realization of the epidemic model. We find the $CFR_{q,t}$ and $IFR_{q,t}$ for each model realization as the number of deaths over observed infections, and number of

deaths over total infections, respectively. Repeating and calculating summary statistics across the 2000 model realizations achieves the 95% CI. This process therefore accounts for the uncertainty in the estimated parameters and stochasticity in the epidemic model, but not from the risk model estimates.

# 3 Results

## 3.1 Model and parameter estimates

**3.1.1 Model fits.** Fig 3 summarizes the epidemic model fit with COVID-19 data for LAC from March 1, 2020 through March 1, 2021 for all disease states across multiple views: New cases, representing new daily incidence; the current number in a compartment at a specific date, relevant for understanding current prevalence rates and comparing with healthcare capacity limitations; and cumulative counts until a specific date. Observed data for available compartments are plotted as black dots, evidencing the day-to-day variability in case and death counts. The figure demonstrates that good model fits are achieved in all compartments across time. Close-ups of the timeseries of the numbers in infection states plotted against available data are provided in Section 3.1 in S1 Appendix.

Section 3.2 in S1 Appendix provides values of the five estimated parameters at two-week intervals from March 1, 2020 through March 1, 2021. The two-step parameter estimation approach (broad grid search to select a single mode of each parameter, followed by approximate Bayesian computation (ABC) using a prior distribution specified around that single mode) achieved convergence in posterior densities. Convergence is not reached for the broad grid search step, with multi-modal distributions returned for each parameter (not shown). By specifying a narrow prior distribution around a mode chosen from the broad grid search sampling, convergence around a dominant single mode is achieved in the final posterior density returned by the ABC sampling step (see Section 3.3 in S1 Appendix for density plots of prior and posterior distributions).

**3.1.2 Epidemic timecourse in LAC.** The LA City Mayor's Office distinguishes between three stages of the COVID-19 epidemic in LA City and County relating to policy response measures implemented following the orders of the County Health Officer: Stage I, March 19—May 7: the initial shutdown; Stage II, May 8—June 11: the first steps towards reopening; Stage III, June 12 and beyond: greater reopening followed by "modifications" closing higher risk settings (including bars and indoor seating in restaurants) [34, 35]. The start of the school year on August 18, although virtual, marked a change in activity level and is also depicted. We characterize three waves of the epidemic occurring across these stages: a first wave, March 1—May 6, 2020, occurring between Stage I and the beginning of Stage II and peaking on April 1; a second and larger wave, May 7—October 14, 2020, beginning with Stage II and peaking on July 30 during Stage III; and a third and more than five times larger wave that began on October 15, 2020 and subsided by March 1 2021. Fig 4b–4f characterize the estimated model parameters relative to these policy stages and epidemic waves; a full time course of the epidemic and policy decisions in LAC can be found at [36].

We estimate that for most of the first wave, which coincides with Stage I, the overall observation rate was $r_t = 0.19$ (95% CI: 0.12,0.26) of all infections observed. Beginning in mid-April 2020, the observation rate began to steadily increase through the second wave and Stages II and III until levelling off at a value of $r_t = 0.5$ (0.34, 0.64) of infections observed by August 15, 2020. In the initial period of the outbreak before public behavior began to change and policy interventions were implemented, we estimate the basic reproduction number in LAC was $R0 = 3.69$ (3.6, 3.82). From March 12 to March 27, 2020, beginning just before the Stage I lockdown was implemented, we estimate a reduction to an $R_t$ of 0.88 (0.77, 0.95). The

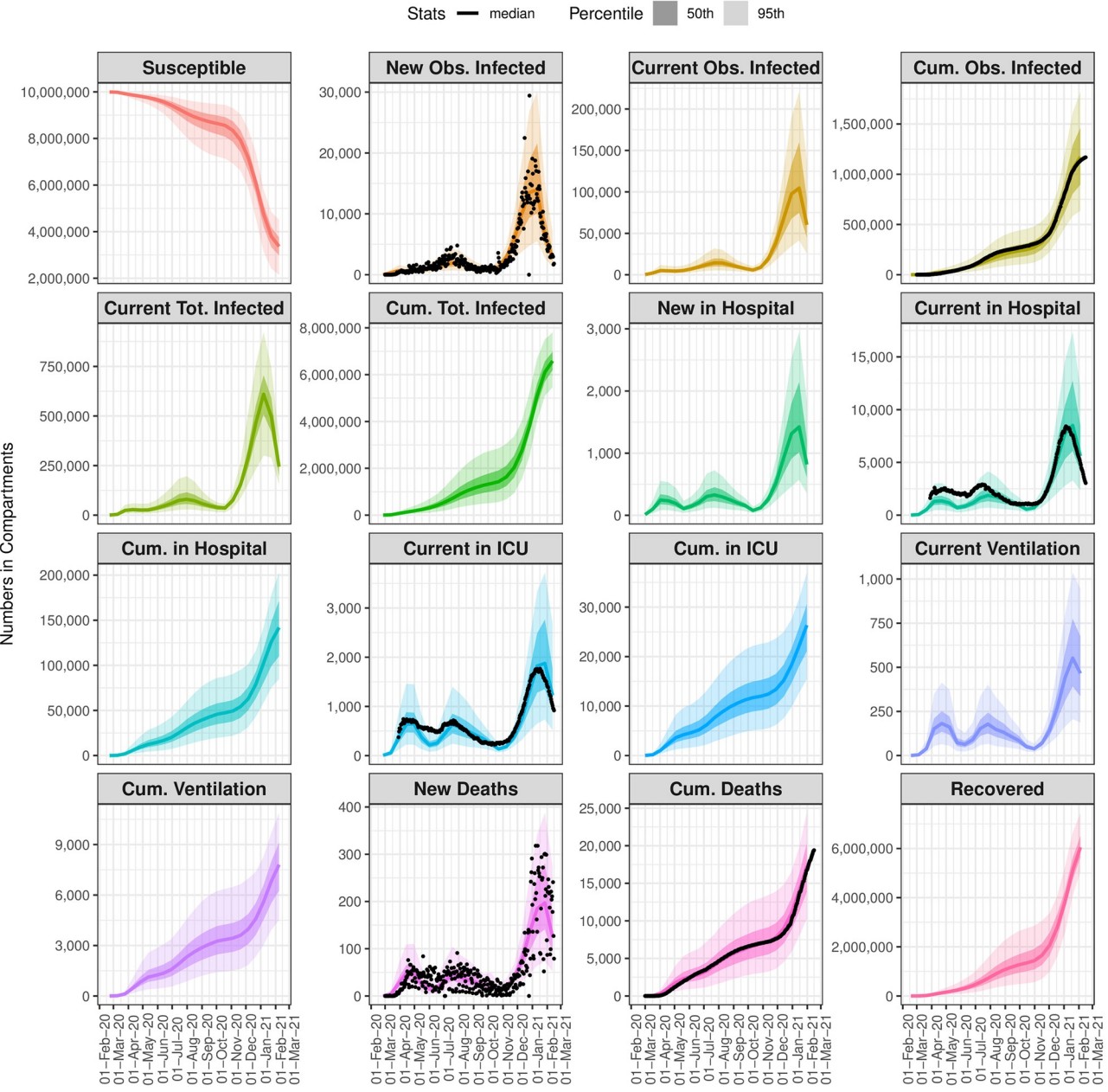

**Fig 3. Summary of the epidemic model fit with COVID-19 data for Los Angeles, for all state variables, across multiple views: New cases, representing new daily incidence, current number in a compartment at a specific date, and cumulative counts.** Available observed data (for new and cumulative counts) are plotted as black dots. Estimates are shown as the median number in compartments over time, with the 50% (darker) and 95% (lighter) CI.

corresponding reduction in transmission led to a levelling off at 36,000 (14,000, 79,000) estimated current total infections (including observed an unobserved) on April 2, 2020.

$R_t$ remained below 1 until the end of April, reaching 1.26 (1.06, 1.39) just as the Stage II reopening began. The increase in $R_t > 1$ facilitated the increase in infections and the second wave of the epidemic, which peaked at 105,000 (41,000, 200,000) current total infections. Following the decrease in the susceptible population due to the sizeable number of cases accrued from the second wave, $R_{t,eff}$ began to diverge appreciably from $R_t$. By mid-July $R_{t,eff}$ had

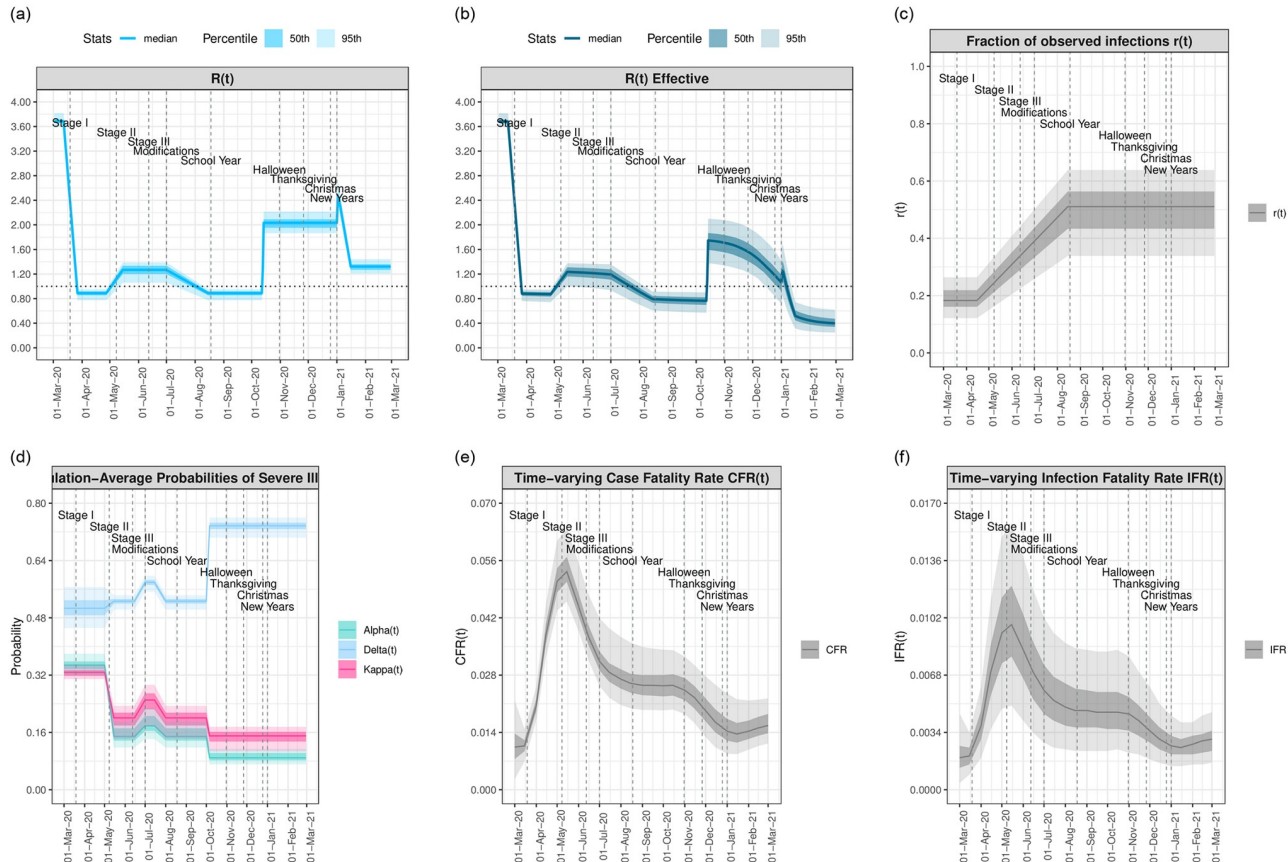

**Fig 4. Timeseries of model-estimated parameters relative to key dates and COVID-19 policy decisions in LAC.** Model-estimated median curves are plotted along with the 50th% (dark shading) and 95% CI (light shading). (**a**) $R_t$, the time varying reproduction number. (**b**) $R_{eff,t}$, the effective reproduction number. (**c**) $r_t$, proportion observed infections. (**d**) $\alpha_t$, the probability of hospitalization given infection; $\kappa_t$, probability of ICU admission given hospitalization; and $\delta_t$, probability of death given admission to the ICU. (**e**) Population-average case fatality rate, $CFR_t$. (**f**) Population-average infection fatality rate, $IFR_t$.

dropped below 1, whereas $R_t$ based on behavior alone took two weeks longer to drop below 1. $R_{t,eff}$ reached a lowest value of $R_{eff,t} = 0.76 (0.58, 0.91)$ in mid-August, just around the time the school year began virtually, where it remained through mid-October.

Following the further re-opening of personal service businesses, malls, and outdoor drinking establishments in late September through October, the third wave began. We estimate an $R_t$ (based on behavior alone) of 2.03 (1.87, 2.22) from October 15, 2020—January 1, 2021, and a final spike of 2.40 (2.15, 2.55) from January 1—January 5, before beginning to decrease to 1.06 (0.97, 1.17) by January 15. Meanwhile, with soaring infections the *effective* $R_{eff,t}$ dropped back below 1 by January 5, 2020, even as the $R_t$ based on behavior alone was spiking at values above 2. $R_{t,eff}$ remained below 1 through March 1, 2021. Current observed infections peaked at approximately 580,000 (275,000, 850,000) on January 13, meaning that at this time over 5% of the LAC population was currently infected, compared with just over 1% of the population with current observed infections. By March 1, 2021, current observed infections had dropped back down to pre-surge levels of October 15, 2020. When unobserved cases are accounted for, we estimate that 40%–60% of the LAC population had been infected with COVID-19 by March 1, 2021.

We identify three phases of the probabilities of disease stage progression, $\alpha_t$, $\kappa_t$, and $\delta_t$ across the three waves. The highest values for all three probabilities were observed during the first wave, which as we will see below was marked by a proportionally higher fraction of infections from the higher-risk elderly population and in particular those 80+, a lot of whom coming from SNFs/care homes; $\alpha_t = 0.349(0.322, 0.38)$, $\kappa_t = 0.327(0.309, 0.348)$, and $\delta_t = 0.506$ (0.452, 0.565). The three probabilities decreased starting at the beginning of May as infections started to increase, this time with a higher proportion coming from younger, lower-risk populations. From mid-May through mid-October, 2020, $\alpha_t = 0.153(0.118,0.19)$, $\delta_t = 0.196(0.143, 0.234)$, and $\kappa_t = 0.526(0.503, 0.543)$, with a brief increase to values 0.184 (0.142, 0.228), 0.246 (0.179, 0.293), and 0.579 (0.553, 0.597), respectively, during the height of the second wave in mid-late July. With increasing infections in the third wave, $\alpha_t$ and $\kappa_t$ further decreased to values 0.1 (0.077, 0.124) and 0.157 (0.114, 0.187), respectively. Meanwhile, $\delta_t$ increased to its highest value yet of 0.737 (0.704, 0.76), suggesting that while the infected population had become either lower risk for hospitalization or less inclined to seek treatment in healthcare, for those making it through to the final stages of hospitalized care, their probability of survival was low. Despite the drop in $\alpha_t$ during the third wave, the surge of infections meant that hospital capacity was surpassed for the first time during this wave (Fig 4 in S1 Appendix). ICU capacity was not surpassed at any point, although it was approached during the third wave.

The population-wide $CFR_t$ and $IFR_t$ are also characterized by three phases, peaking at the beginning of May 2020 before the probabilities of disease progression began to drop, leveling out through the summer of 2020, and decreasing further with the third wave as the initial influx of cases outpaced the deaths, which later started to catch up. On May 15, 2020, marking the majority of deaths that could have come from the end of the first epidemic wave, $CFR_t = 5.56\%(4.35\%, 6.3\%)$ and $IFR_t = 1.1\%(0.41\%, 1.81\%)$; on October 15, 2020, marking the majority of deaths that could have come from the end of the second epidemic wave, $CFR_t = 2.74$ (2.08, 3.39) and $IFR_t = 0.55(0.22, 0.96)$; and on March 1, 2021, marking the majority of deaths that could have come from the end of the third wave, $CFR_t = 1.65(1.29, 2.06)$ and $IFR_t = 0.32$ (0.16, 0.55). Here we have provided values at the end of each wave to allow the number of deaths occurred from each wave to catch up with the number of infections.

## 3.2 Conditional relative risks (RR) for risk factors

Table 1 displays as mean and 95% CI the marginal relative risks (RR) extracted from the literature (left column) and the RR for BMI, smoking, and comorbidity conditional on age estimated by JAM for the three risk models representing increasing disease progression: hospitalization given infection, $(H|I)$, ICU admission given hospitalization, $(Q|H)$, and death given ICU admission, $(D|Q)$. We observe that the independent effect of comorbidities and obesity attenuate with increasing severity of disease; smoking may increase with age, however a very wide confidence interval for $(H|I)$ makes this conclusion tentative.

Separately, the estimated conditional RR for each age group, estimated such that the distribution of age groups in the infected and deceased populations produced by the model matches that observed in LAC, are shown in Table 2. Our modeling approach estimates that the conditional risk of advancing to the next stage of illness relative to the reference population is equivalent across the three models, i.e. $(H|I) = (Q|H) = (D|Q)$; we report this single set of values for each age category in Fig 2. The independent effect of age is much higher than from any other risk factor, and increases exponentially with age. Fig 11 in S1 Appendix shows the frequency of the age groups in the deceased population estimated by the model compared with the observed frequency, i.e. the distributions featured in the objective function used to estimate the conditional RR for the age groups for the three models in Step 5 of the risk model methodology.

**Table 2. The estimated conditional relative risk (RR) for each age group relative to the reference age group of 19– 49.** The estimated conditional RR of advancing to the next stage of illness is equivalent across the three models, $(H|I)$, $(Q|H)$, and $(D|Q)$. The RR are conditional to the other risk factors and estimated from LAC infection and death data stratified by age, using the combination of the epidemic and risk models.

| Age group | $(H|I) = (Q|H) = (D|Q)$ |
|---|---|
| $0 - 18$ | 0.14 |
| $19 - 49$ | 1 |
| $50 - 64$ | 2.59 |
| $65 - 79$ | 6.69 |
| $80+$ | 18.17 |

### 3.3 Risk-profile-stratified probabilities of disease stage progression

Tables 4, 9–11 in S1 Appendix show the estimated probabilities of disease stage progression for each of three models $P_t(\widehat{H|I})$, $P_t(\widehat{Q|H})$, and $P_t(\widehat{D|Q})$, as well as the estimated frequency in the overall LAC population stratified across each risk profile characterizing a unique combination of age group, BMI range, smoking status, and any comorbidity. Profile-stratified probabilities are shown for dates ranging every two weeks from May 15 2020—March 1 2021, The probability of hospitalization given infection, ICU admission given hospitalization, and death given ICU admission vary extensively across the risk profiles. Notably, the risks within specific marginal factor groups also vary extensively. To demonstrate the variability in disease stage progression probability across risk profiles, we show in Fig 5a–5c the range of values that can be taken on by profiles falling within each age group. Specifically, these figures show the mean (as a point), and minimum and maximum (error bar) of the probabilities across the composing risk profiles within each age group $0 - 18$, $19 - 49$, $50 - 64$, $65 - 79$, 80+, under each of the three risk models. The figures demonstrate that the difference in probability between profiles within an age group may vary more widely than across adjacent age groups, in particular for the probability of hospitalization given infection $P_t(\widehat{H|I})$; the factor increase between adjacent age groups ranges from 1.25 to 2 while the ratio increase from the maximum to minimum profile ranges from a factor of two to four times.

### 3.4 Risk-profile-stratified $CFR_{q,t}$ and $IFR_{q,t}$

Tables 12 and 13 in S1 Appendix show the median and the 95% CI of the estimated risk-profile-stratified case fatality rates $CFR_{q,t}$ and infection fatality rates $IFR_{q,t}$ across dates every two weeks from May 15 2020—March 1 2021. To facilitate interpretation of the variability in these quantities across risk profiles, we show in Fig 6a and 6b the range of values that can be taken on by profiles falling within each age group; the mean (point), and minimum and maximum (error bar) of the *median $CFR_{q,t}$* and $IFR_{q,t}$ across the composing risk profiles within each age group are shown. The maximum $IFR_{q,t}$ for each age group comes from individuals with at least one comorbidity, a smoking history, and severe obesity, while the minimum comes from individuals that have no comorbidities, do not smoke, and have a healthy BMI.

　　As with the probabilities of disease progression, the $CFR_{q,t}$ and $IFR_{q,t}$ vary extensively both across age groups, as well as across profiles representing different combinations of risk factors for a given age group. The factor differential between the risk profiles decreases with age, varying from a 3- to 30-fold difference across profiles within each age groups. There are also changes in the $CFR_{q,t}$ and $IFR_{q,t}$ across epidemic waves for the higher age groups. On May 15, 2020, marking the end of deaths from the first wave, median $IFR_{q,t}$ ranged from 0.01% (0.006%, 0.019%) to 0.27% (0.14%, 0.44%) across profiles for ages $19 - 49$; on March 1, 2021,

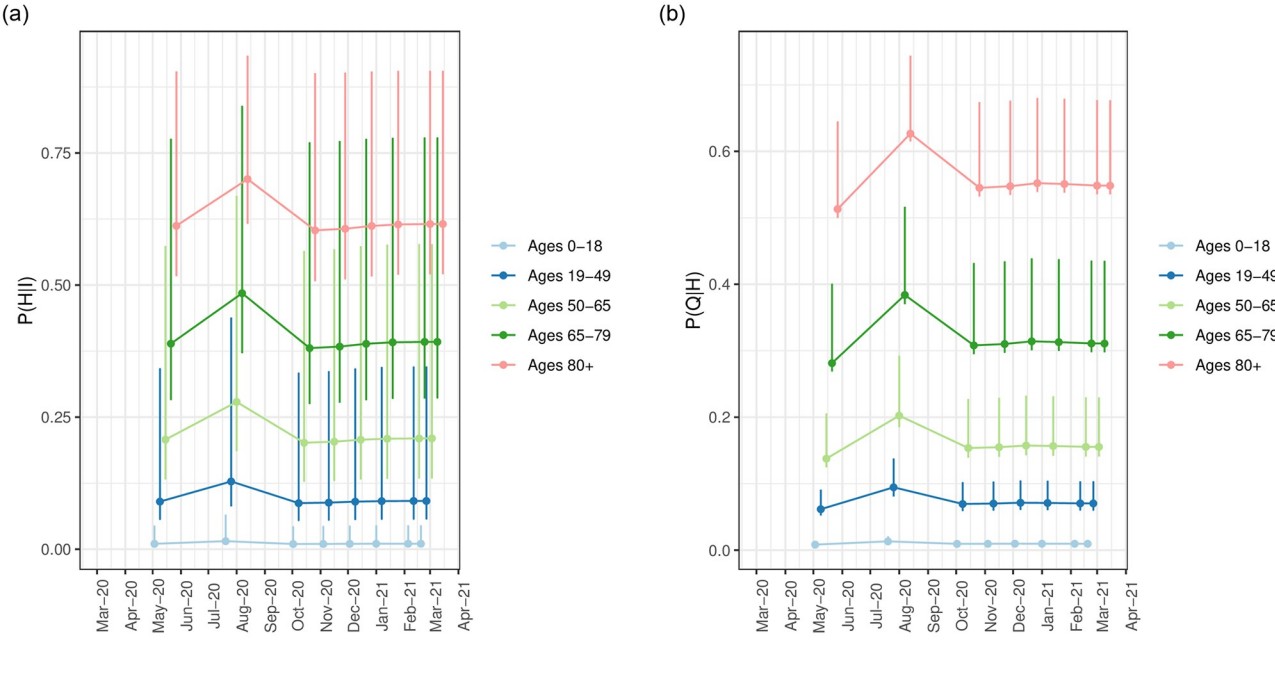

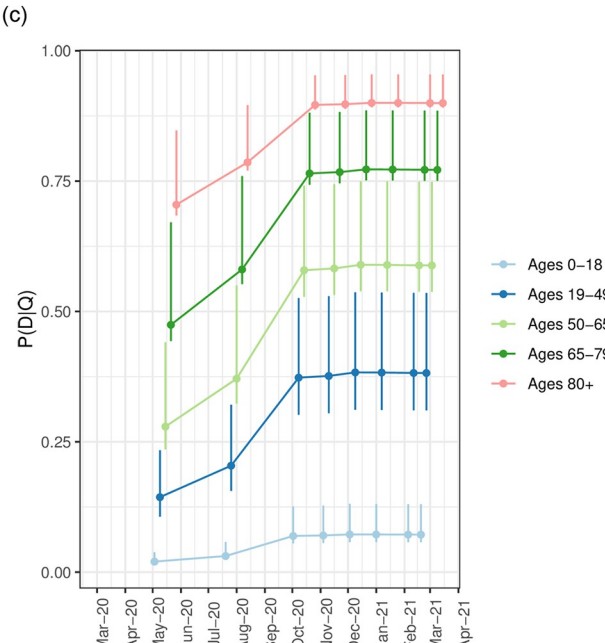

**Fig 5. Range of the probabilities of hospitalization given infection across each risk profile, $P_t(\widehat{H|I})$, summarized for each age group.** Each figure shows the mean (as a point), and minimum and maximum (error bar) probability for each age group $0-18, 19-49, 50-64, 65-79, 80+$, under each risk model. (**a**) Range of the probabilities of ICU admission given hospitalization across each risk profile, $P_t(\widehat{Q|H})$, summarized for each age group. (**b**) Range of the probabilities of death given ICU admission across each risk profile, $P_t(\widehat{D|Q})$, summarized for each age group. (**c**) Range of probabilities of disease stage progression for the three models $P_t(\widehat{H|I})$, $P_t(\widehat{Q|H})$, and $P_t(\widehat{D|Q})$ across all risk profiles within each age group.

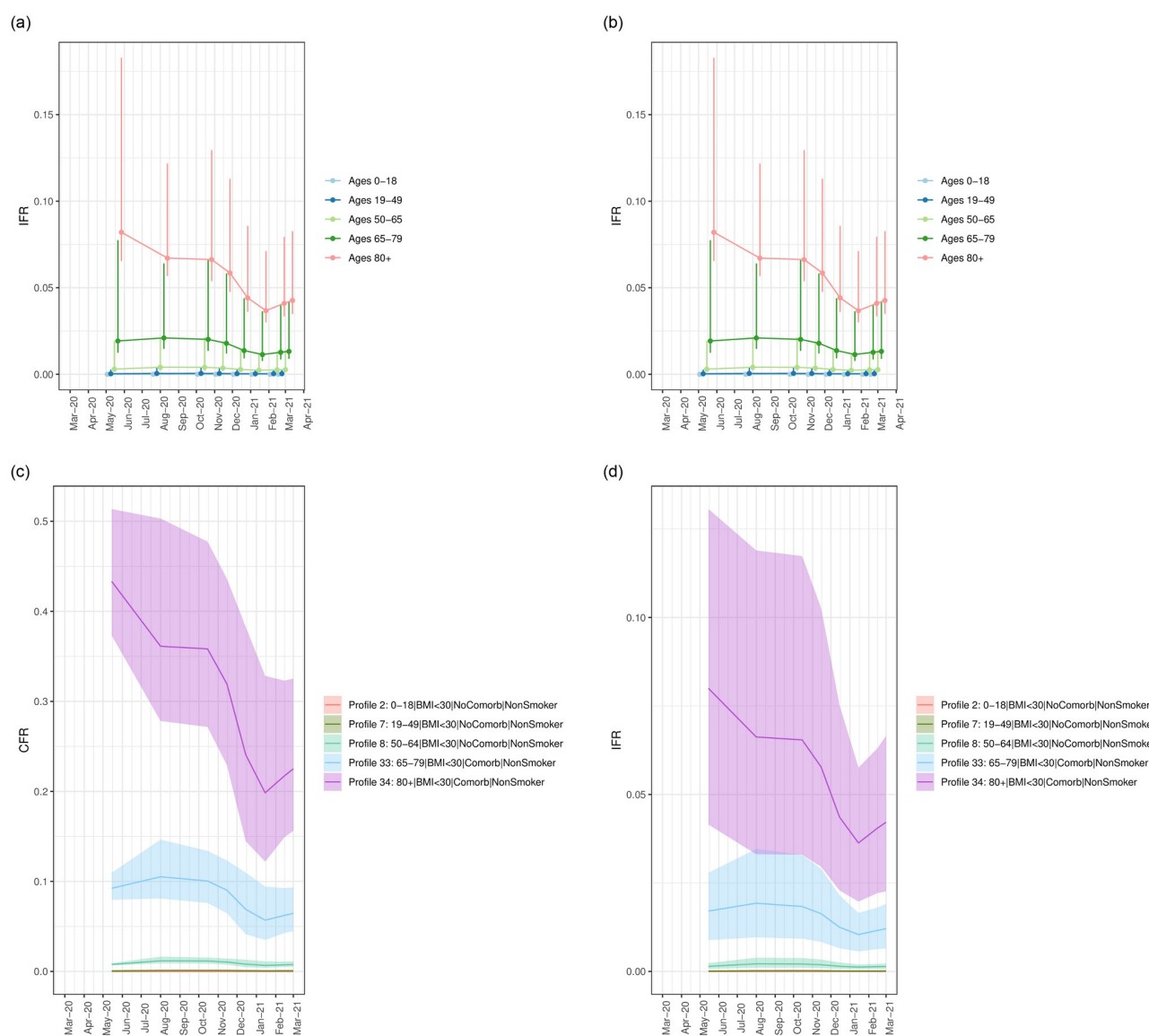

**Fig 6. a-b**: Range of $CFR_{q,t}$ and $IFR_{q,t}$ values taken on by risk profiles within each age group. Each figure shows the mean (as a point), and minimum and maximum (error bar) of the median $CFR_{q,t}$ and $IFR_{q,t}$ for each age group $0 - 18$, $19 - 49$, $50 - 64$, $65 - 79$, $80+$. **c-d**: Median (line) and 95% CI (shading) of the $CFR_{q,t}$ and $IFR_{q,t}$ of the most-populous risk profiles for each age group. (**a**) Range of $CFR_{q,t}$ taken on by risk profiles within each age group. (**b**): Range of $IFR_{q,t}$ taken on by risk profiles within each age group. (**c**) $CFR_{q,t}$ of the most-populous risk profile for each age group. (**d**) $IFR_{q,t}$ of the most-populous risk profile for each age group.

marking the end of deaths from the third wave, the $IFR_{q,t}$ for $19 - 49$ remained the same. For ages $50 - 64$ on May 15, 2020, median $IFR_{q,t}$ ranged from 0.14% (0.075%, 0.23%) to 1.9% (1.0%, 3.14%); these values remained approximately constant through March 1, 2021. For ages $65 - 79$ on May 15, 2020, median $IFR_{q,t}$ ranged from 1.25% (0.65%, 2.04%) to 7.7% (4.02%, 12.6%); the minimum and maximum median $IFR_{q,t}$ values decreased to 0.90% (0.48%, 1.41%) and 4.22% (2.23%, 6.65%), respectively, by March 1, 2021. For ages 80+ on May 15, 2020, the $IFR_{q,t}$ ranged from 6.5% (3.4%, 10.7%) to 18.3% (9.5%, 29.9%) but by March 1, 2021, had dropped to 3.5% (1.9%, 5.5%) and 8.3% (4.4%, 13.3%), respectively.

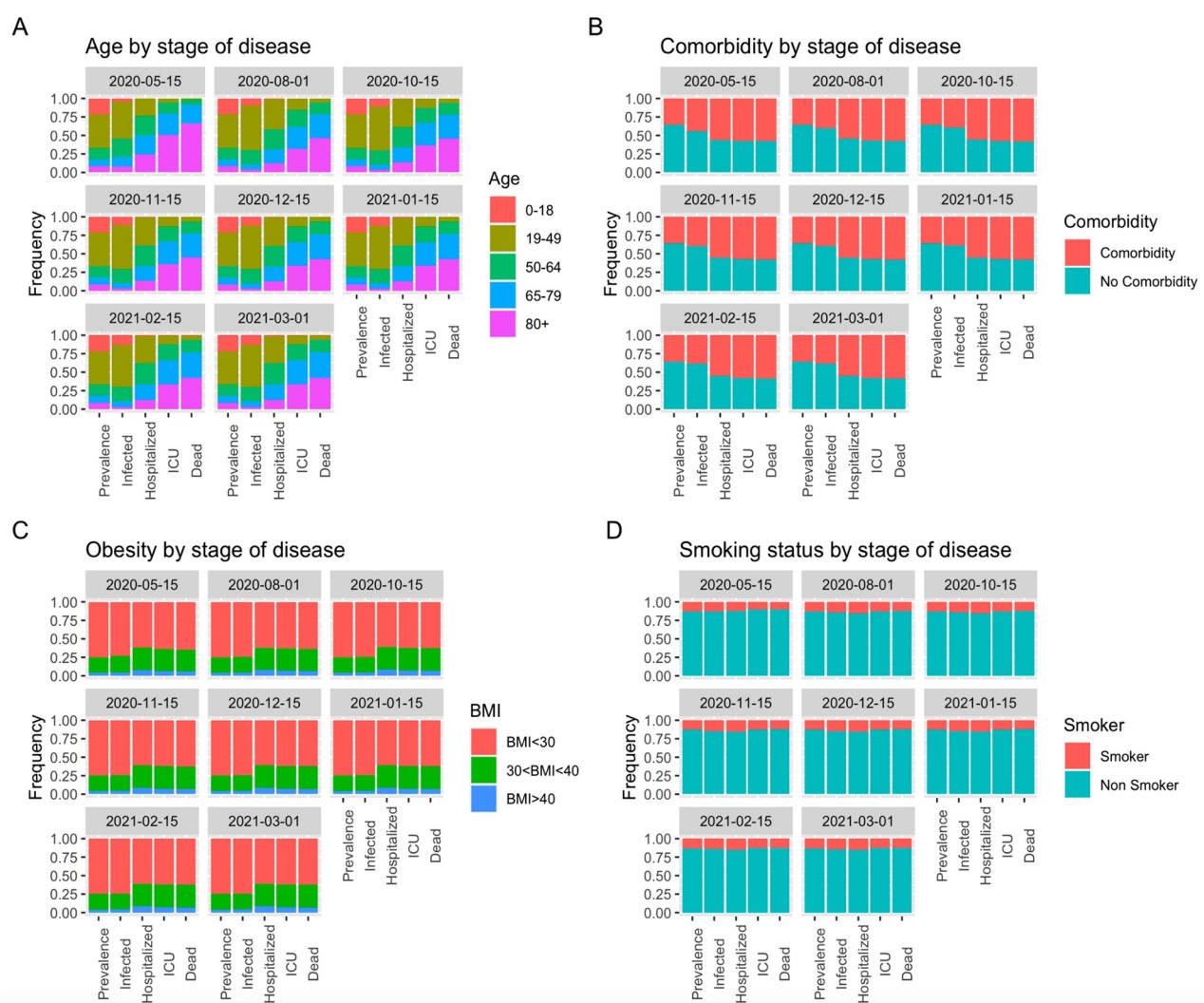

**Fig 7. Estimated frequency of risk factor groups in the overall LAC population, and the distribution of individuals in each stage of disease from infected, hospitalized, admitted to ICU, to deceased at two week intervals from May 15, 2020 through March 1, 2021.** (A) age group, (B) any comorbidity group, (C) obesity class group, and (D) smoking status group.

We also plot the estimated median and the 95% CI for the most prevalent risk profile within each age group in Fig 6c and 6d, to compare with the extremes presented in Fig 6a and 6b. The CI for the $IFR_{q,t}$ are wider than for the $CFR_{q,t}$, since the former account for uncertainty in the infection observation rate, $r_t$. Although wide, the CI are non-overlapping between the most-prevalent profiles for each age group visualized in the figures.

### 3.5 Frequency of risk factor groups at each stage of disease

Fig 7 shows the model estimated frequency of risk factor groups (age groups, obesity class groups, any comorbidity, and smoking) in the population of individuals in each stage of disease (infected, hospitalized, admitted to ICU, and deceased), compared with their frequency in the overall LAC population at two week intervals from May 15, 2020 through March 1, 2021.

The figure illustrates the effect of each risk factor on its frequency in the population at each stage of disease. Age greater than that of the reference of 19 – 49 years has the largest effect on

each probability of disease progression and the most apparent increase in frequency at each stage of disease. Both comorbidities and obesity have a much larger effect on the probability of hospitalization given infection than death given hospitalization, which corresponds to a large increase in the proportion of individuals with comorbidities and obesity in the infected population and hospitalized populations, but little difference between the hospitalized, in-ICU, and deceased population. Smoking shows the smallest effect overall out of all risk factors, in part because this risk factor is infrequently represented in the risk profiles more prevalent in the infected population.

Most notably, Fig 7a makes apparent the decrease in the frequency of higher age groups in the infected population between the first and third epidemic waves, and the effect this change had on the composition of the hospitalized, in-ICU, and deceased populations. For example, individuals 80+ show a steep increase in the fraction of the population progressing to each stage of disease; between May 15, 2020 (marking the end of deaths that could have come from the end of the first epidemic wave), and March 1, 2021 (marking the end of deaths that could have come from the third wave), the observed frequency of individuals 80+ in the infected population decreased from 10% to 3% and the prevalence of this group in the hospitalized population changed from 25% to 12.5%, in the in-ICU population from 50% to 35%, and in the deceased population from 65% to 40%. Meanwhile, individuals aged 19 − 49 show a steep decrease in the fraction of the population as disease stage progresses; increasing from 50% to 62% of the infected population, 24% to 36% of the hospitalized population, 7% to 12% of the in-ICU population, and 2% to 6% of the deceased population between May 15, 2020, and March 1, 2021.

## 4 Discussion

This work has developed a framework for using available data on COVID-19 epidemic dynamics and prevalences of COVID-19 risk factors at the population level to estimate time-varying subpopulation-stratified probabilities of disease progression and *CFR* and *IFR* during three epidemic waves in Los Angeles County from March 1, 2020, through March 1, 2021. In the absence of individual-level data, the technical contribution of this work was to integrate a dynamic epidemic model with a risk modeling approach to estimate conditional effects from available marginal data and to subsequently produce time-varying subpopulation-stratified estimates for LAC. To reflect the uncertain knowledge of many parameters and the understanding that in non-linear systems small variations to specific parameters can result in large impacts in outputs [37], we account for uncertainty in all results through the use of a stochastic epidemic model and a Bayesian approach to parameter estimation. The epidemic modeling framework produces estimates with confidence intervals of the population-wide reproductive number, case observation fraction, probabilities of disease progression, and *CFR* and *IFR*. On its own, the risk model estimates the conditional effects of each risk factor and therefore the overall effect of risk factors in combination. These adjusted effects have not been typically reported in observational studies on COVID-19, yet help to understand more precisely what subpopulations are at highest risk of advancing to each stage of disease. Integration of the risk model with the epidemic model allows the comparison of dynamic outcomes and parameters across the overall population, age groups, and more fine-grained subpopulations in LAC representing age and combinations of other risk factors for severe COVID-19 illness. Such fine-grained results can be useful in understanding disparities in the effect of the epidemic on different groups in LAC, and can inform studies involving targeted subpopulation-level policy interventions [16].

We focus our modeling framework on the risk factors age, comorbidities, obesity, and smoking status as these demographic and medical conditions have consistently been identified across various studies as factors inducing the probability of progressing to severe illness given COVID-19 infection [8]. We do not include race/ethnicity as a factor, because although strongly predictive of the risk of overall mortality from COVID-19 [17], it has been shown that increased exposure risk and not race *per se* explains racial disparities in COVID-19 health outcomes [38, 39].

Analyses demonstrate that the risk of severe illness and death from COVID-19 infection have decreased over time and moreover vary tremendously across subpopulations representing combinations of the four modeled risk factors, which we call *risk profiles*, suggesting that it is inappropriate to summarize epidemiological parameters for the entire population and epidemic time period. This includes variation not only across age groups, but also within age strata combined with other risk factors analyzed in this study. The highest *IFR* for each age strata come from profiles including comorbidities, obesity Class 2 or 3, and current smoking status. The factor differential between the risk profiles with highest and lowest *IFR* within each age strata decreases with age. At the end of the first epidemic wave, we find median *IFR* ranging from 0.01% to 0.27% across risk profiles for the age group $19 - 49$, an almost 30-fold difference; ranging from 0.14% to 1.9% across profiles within age group $50 - 64$, a 14-fold difference; from 1.25% to 7.7% for ages $65 - 79$, a 6 fold difference; for ages 80+, the range was from 6.5% to 18.3%, a 3-fold difference.

Our age-stratified *IFR* estimates during dates corresponding to the first and second epidemic waves in LAC (May—October, 2020) are comparable to those found in recent notable reviews and modeling studies including a meta-regression of seroprevalence data from 11 European countries and 12 U.S. locations [8], a study comparing mortality data from 45 countries with 22 seroprevalence studies [40], and a model-based analysis for estimating *IFR* during in New York City's large first epidemic wave (March—May, 2020) [10].

A feature of our *IFR* estimates for the higher age groups (65+) is that they decreased in the third epidemic wave; median *IFR* for ages $65 - 79$ ranged from 1.25% to 7.7% after the first wave compared with 0.90% to 4.22% after the second wave; for ages 80+ the median *IFR* ranged from 6.5% to 18.3% after the first wave and had dropped to a range of 3.5% to 8.3% after the third wave. The decrease in *IFR* for the same profiles during the third wave may be explained by three factors. First, the first and second waves were characterized by a large number of outbreaks in nursing homes/SNFs; it has been demonstrated that when high rates of infection have occurred among nursing home residents, *IFRs* for the same age group (and the overall population-average *IFR*) will be significantly greater then when cases in care-home-aged populations have been in the general community due to greater frailty in care home populations [40]. Second, there may have been improvements in medical treatment over the course of the epidemic [41, 42]. Third, a limitation of our model-based analysis for older risk strata is that we assume unobserved infections are equally distributed across all risk profiles, whereas there are likely to be far fewer unobserved or asymptomatic infections for those at higher risk of severe outcomes. For risk profiles including individuals age 65+, and for dates during the third wave when the number of infections spiked especially among younger age groups, our *IFR* may therefore be underestimated and the true values lie between *IFR* and *CFR* estimates. For the most prevalent profile within age group $65 - 79$, the *CFR* was 6.2% (95% CI: 4.2%, 9.3%), and for the most prevalent profile within age group 80+, the *CFR* was 21.7% (14.9%, 32.3%). More generally, in interpreting our results for policy implications, emphasis should be placed on the relative differences in *IFR* across risk profiles and the understanding that the *IFR* for a specific age strata represents an average across a wide variation given the presence or absence of other risk factors.

Our overall *IFR* estimate for LAC at the end of the first epidemic wave of 1.11% (95% CI: 0.41%, 1.81%) is similar to the overall *IFR* estimated in the NYC study after the first large wave, when only confirmed deaths are accounted for, of 1.10% [10]. *IFR* estimates at the end of the second wave are equivalent to the global estimate of 0.5% as of September, 2020 coming from the study by O'Driscoll et al. (2020) utilizing mortality and seroprevalence data across 45 countries [40]. The decrease in *IFR* between the first and second waves follows from the decrease in the prevalence of populations aged 65+ in observed infections from approximately 23% on April 15, 2020 to 12% and lower as of July 15, 2020 [14], but may also reflect other changes in the demographic composition of infected individuals including other at-risk sub-populations for which stratified data in LAC is not available (e.g., individuals with comorbidities). The slight decrease in overall *IFR* at the end of the third wave of 0.32% (0.16%, 0.55%) may reflect the decrease in age groups 65+ in the observed infected population from 12% on July 15, 2020, to 10% on December 12, 2020, and a proportional increase in unobserved infections from younger age groups.

Our estimates may misrepresent the true *IFR* from COVID-19 in LAC because we account only for underascertainment of infections and not of deaths [43, 44]. Although we assume that the underascertainment of deaths is much lower than for infections, the percentage is likely to have been the highest during the third wave given that the percentage of documented at-home deaths increased from 4% in the first wave to 9% in the third wave (E. Garcia, personal communication based on unpublished data from the state of California, April 20, 2021); this may most affect estimates for older age groups in which unaccounted for deaths are likely to be the highest [40, 45]. Even at the lowest overall *IFR* estimated for LAC, a key finding is that COVID-19 is substantially more deadly than seasonal influenza, which has a population-average *IFR* of approximately 0.05% [8, 46].

A critical factor determining our *IFR* estimates is the fraction of cases that are detected, $r(t)$. We estimate that this was 19% (95% CI: 12%, 26%) in the first wave of the outbreak and had stabilized to 50% (34%, 64%) in the late summer through the fall, following the peak of the second wave. There is insufficient serological information for LAC to provide confirmatory evidence behind these estimates, and CDC studies of serology carried out in various settings throughout the USA (not including LAC) during the first epidemic wave vary from as low as 2.3% of infections observed to as high as 30% of infections detected [26]. An additional piece of evidence that supports low detection rates is the low fraction of infections that seek medical attention, since this informs how many infections are of at least moderate COVID. A recent study has used serological studies, participatory surveillance systems, and mathematical modeling to estimate the underdetection of infections in France and found that only 31% of individuals with COVID-19-like symptoms consulted a doctor in the study period. Although a different context, this result suggests that large numbers of symptomatic COVID-19 cases do not seek medical advice and therefore many of these likely do not show up in the official register of cases [47].

This study is prone to typical limitations occurring when modeling epidemiological dynamics in the context of a rapidly evolving infectious disease outbreak. We model the major epidemic trends across the three waves using time-varying parameters, however this approach does not enable capturing all of the complexity of the changing epidemic. Due to the model specification and the concurrent estimation of multiple time-varying parameters, multiple joint parameter solutions exist, resulting in multimodal posterior distributions. We attempted to address this by employing a two-step parameter estimation approach, first using broad grid search to identify and choose a mode of a posterior distribution, and second using approximate Bayesian computation to identify the shape of the distribution around the chosen mode. This process involved the use of "expert opinion" to guide the choice of parameters towards

most correctly representing the major trends and peaks at each epidemic wave, at the expense of accurately capturing deviations from the major trendlines.

Data informing the conditional effect estimates within the risk model were therefore aggregated across early, large, retrospective studies from China (for comorbidities and smoking) [3] and NYC (for BMI) [2] on the fractions of hospitalization, ICU admission, and death by individual risk factors. We chose these studies due to the limited body of research reporting marginal or conditional risk effects for the same cohort across the three modeled stages of disease progression, returned by a Pubmed search at the start of this study. While we attempt to reframe these results for the demographic composition of the LAC population through regional data on the prevalence of risk factors and the correlation structure between risk factors, there may be differences in the underlying study population or treatment setting between China, NYC, and LAC that would lead to heterogeneity in effect estimates. However, we believe that the estimates from the Chinese studies do represent population-based estimates as these samples avoid some of the biases present from other potentially available studies, but with highly selected samples.

While this work has focused on demonstrating the substantial heterogeneity in risk probabilities and *IFR* across subpopulations, it employs a single-population epidemic model. LAC is a large county consisting of many composing cities and communities, each with their own epidemic processes unfolding at different rates [48]. Extreme disparities in infection incidence and mortality have been observed for different communities within LAC. This includes incidence rates up to 15 times higher in low-income neighborhoods in East LA with high percentages of essential workers than in affluent communities in West LA [14], and COVID deaths as a proportion of the typical total deaths 11.6 times higher for young, foreign-born Latinx than for young, U.S.-born, non-Hispanics (for California) [17]. These large differences in infection incidence and death will undoubtedly translate into large differences in probabilities of disease progression and *IFR*. However, at the time of beginning this study we did not have the data to formally model subpopulation-specific probabilities of exposure or the data on hospitalization and death counts for different groups necessary to fit the parameters of a multi-population model. The approach we developed is a way to use commonly available population-level epidemic timeseries data to model multiple groups in a single population, and combine these population-level estimates with prevalence rates of risk factors to produce stratified estimates for different subpopulations, specific to the region of LAC. This adaptive approach allowed us to provide epidemic trends and risk estimates that informed the LAC Department of Public Health and other decision-makers in real-time during the emerging epidemic.

Future work will develop multi-population models that estimate subpopulation-stratified probabilities of infection, of illness progression, and *IFR*, accounting for key risk factors of both exposure to infection and severe illness given infection. Risk factors for exposure are not limited to age and health conditions, but also include more diverse socioeconomic factors including occupation and essential worker status, neighborhood of residence, housing overcrowding, multigenerational households, economic status, and access to PPE [48–52]. In the meantime, the subpopulation-stratified estimates of disease progression and *IFR* produced using the framework presented here can be used to evaluate policy decisions that may involve both population-wide interventions and interventions that target specific subpopulations at risk of developing severe illness given infection, for example isolating or prioritizing vaccination for the elderly or those with other health-related risk factors [16].

## Supporting information

**S1 Appendix.**
(PDF)

## Acknowledgments

We would like to acknowledge data processing and visualization support from Claire Jacquillat.

## Author Contributions

**Conceptualization:** Abigail L. Horn, Wendy Cozen, David V. Conti.

**Data curation:** Faith Washburn, William Nicholas.

**Formal analysis:** Abigail L. Horn, Lai Jiang, David V. Conti.

**Investigation:** Abigail L. Horn, David V. Conti.

**Methodology:** Abigail L. Horn, Lai Jiang, David V. Conti.

**Project administration:** Wendy Cozen, David V. Conti.

**Resources:** Faith Washburn, William Nicholas, Wendy Cozen.

**Software:** Abigail L. Horn.

**Supervision:** Kayla de la Haye, Paul Simon, Maryann Pentz, Neeraj Sood, David V. Conti.

**Visualization:** Abigail L. Horn, Emil Hvitfeldt.

**Writing – original draft:** Abigail L. Horn.

**Writing – review & editing:** Abigail L. Horn, Kayla de la Haye, Paul Simon, Neeraj Sood, David V. Conti.

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
