## [Decision Letter · Decision Letter 0]

18 Feb 2021

PONE-D-21-00882

An integrated risk and epidemiological model to estimate risk-stratified COVID-19 outcomes and policy implications for Los Angeles County

PLOS ONE

Dear Dr. Horn,

Thank you for submitting your manuscript to PLOS ONE. After careful consideration, we feel that it has merit but does not fully meet PLOS ONE’s publication criteria as it currently stands. Therefore, we invite you to submit a revised version of the manuscript that addresses the points raised during the review process.

We look forward to receiving your revised manuscript.

Kind regards,

Martial L Ndeffo Mbah, Ph.D

Academic Editor

PLOS ONE

Journal Requirements:

2. Please include captions for your Supporting Information files at the end of your manuscript, and update any in-text citations to match accordingly. Please see our Supporting Information guidelines for more information: http://journals.plos.org/plosone/s/supporting-information

Additional Editor Comments:

Please, thoroughly address reviewers comments, especially reviewer #2. These should greatly improve the readability of the manuscript and the quality of the manuscript.

Reviewers' comments:

Reviewer's Responses to Questions

**Comments to the Author**

1. Is the manuscript technically sound, and do the data support the conclusions?

Reviewer #1: Yes

Reviewer #2: Partly

2. Has the statistical analysis been performed appropriately and rigorously? 

Reviewer #1: Yes

Reviewer #2: Yes

3. Have the authors made all data underlying the findings in their manuscript fully available?

Reviewer #1: Yes

Reviewer #2: Yes

4. Is the manuscript presented in an intelligible fashion and written in standard English?

Reviewer #1: Yes

Reviewer #2: Yes

5. Review Comments to the Author

Reviewer #1: The manuscript was a pleasure to read, and the supplement was also well organized and helped me understand some of the statements made in the manuscript. Unfortunately I have not stayed up to date on the clinical epidemiology or policy issues surrounding COVID-19 so I will only be able to make comments on technical points.

Substantive comments:

1a. Line 72: Can the authors please describe why there is no A->D transition? I imagine COVID diagnosis obtained only through autopsy is not common, but some explanation here would help.

1b. It may be worth stating upfront that the binary classification of infections as A or I only represents detection/entering the health system, and not asymptomatic/symptomatic, as in other modeling studies.

2. Line 77: I would appreciate some more explanation of what "proxy with error" means, with respect to the Q->V transition. In particular, what real process when a patient enters the ICU does that compartment represent? Does it mean that someone is only recorded as an ICU COVID patient when they are given a ventilator?

3. Section 2.1.1: if beta(t) is the parameter being estimated by ABC, what is the relationship of that estimated curve to mu(t) in the previous paragraph (starting on line 90)?

4. Line 124: I would appreciate some comments from the authors on convergence (or not) of the ABC sampling for the 6 parameters in the main text. If time permits, perhaps trace plots or histograms in the appendix would be a nice addition.

5. Line 161: For those of us unfamiliar with JAM, a sentence describing the assumptions needed to combine the correlation structure and marginal effects to get conditional effects would be a valuable addition.

6. Line 187: I understand that this accounts for uncertainty in estimated parameters and stochastic variation in trajectories, but does it also account for uncertainty in the estimates from the risk model?

7. Line 192: How does changing R(t) adjust beta(t) and mu(t)? I'm still somewhat unclear on the relationship of these quantities.

Minor comments:

1. Abstract: CFR is used before it is defined

2. Line 70: please remind readers which compartments S,E,R are.

3. Line 105: there is an inconsistent use of subscript t and function of t for parameters alpha, kappa, delta between text and Figure 1. Also, p_v is not defined anywhere, what is it?

4. Line 204: text references Fig 4a but looks like it should say Fig 4b.

5. Line 228: text references Fig 4b but looks like it should say Fig 4a.

6. Tables 2 and 3 are very hard to read (too small), could they be enlarged somewhat?

Reviewer #2: In this study, the authors present an analysis of the COVID-19 transmission dynamics in Los Angeles county (LAC). They aim to estimate the probability of severe illness depending on the risk profile of the individuals, and analyse the impact different control measures could have had on the number of infected individuals. In order to carry out this analysis, the authors developed the following workflow:

1 - Estimate the population-wide proportion of hospitalised cases, proportion of hospitalisations leading to Intensive Care Unit (ICU) admission, proportion of deaths given ICU admission, proportion of reported cases, and reproduction number over time using a compartmental model and Approximate Bayesian Computation.

2 - Estimate the conditional relative risk of different factors on the proportion of hospitalisations, ICU admissions and deaths using marginal relative risks from the previous literature and Joint Analysis of Marginal summary statistics (JAM).

3 - Estimate the proportion of cases from different risk profiles using the marginal risk factor in LAC, and the age distribution the reported cases.

4 - Use the population-wide estimates from step 1, the conditional risk estimates from step 2 and the distribution of the risks profiles of the infected population from step 3 to deduce the risks of hospitalisation given infection, ICU admission given hospitalisation, and death given ICU admission for each risk profile.

5 - Generate simulations of outbreaks using different scenarios of Non-Pharmaceutical Interventions.

This paper mixes different complex methods and use different publicly available data sources. I appreciate the time and effort the authors have spent to provide the code and make their analysis reproducible on a Github repository. I also want to highlight the thorough Appendix detailing the approach used in every step of the analysis.

Overall, I believe this is an interesting piece, which can provide important contributions to the field. Nevertheless, I think there needs to be some clarification on some stages of the study. It took me time to understand the justifications behind each step of the workflow, notably what outputs were needed for the final results, and the uncertainty of some of the estimated parameters.

Before getting to major and minor points, I had an overarching comment on the paper:

Because of the number of different stages and models developed in the study, I think the message / workflow sometimes gets lost, or at least I got confused a number of times. I wonder whether the authors would consider splitting this analysis into two papers: One focused on estimating the distribution of risk profiles in infected cases and the probabilities of hospitalisation (ICU and death) associated with each profile (ie steps 2 to 4, and the first objective of this paper), and one focused on the simulations of different scenarios of NPIs and vaccine coverage using the risks profiles (mostly Step 5). This way, the different scenarios of NPI included in the simulations could be deepened and more realistic, and the authors could give more information in the Main Paper on the JAM method and logistic regression they implemented (and use the prior distribution on alpha, kappa and delta for their estimations inn Step 4). I do not think this is a requirement for publication, but I believe this would make it easier for the reader to follow the arguments the authors are presenting.

Major points

1/ Summarise the overall workflow in a Figure.

In the summary of my review, I tried to summarise the workflow the authors implemented from the Main text and Appendix. Although I hope I understood it correctly, it took me time and a few read-throughs to figure out how and why the authors went from one stage to another. I think the authors should add a figure summarising each stage of the analysis, along with their input and output, and how they connect to one another. I believe this would be very helpful for the readers, and would prevent a lot of confusion.

Along with that, the authors use a lot of different notations in each section of the Appendix. I believe they should summarise all the notations of Section 2 in a table (similar to what they did in Table 1, 2, and 3 of the Appendix). This would facilitate the reading and general understanding of their analysis.

2/ The implementation of the compartmental model should be clearer

I am not sure I understand why Approximate Bayesian Computation (ABC) was needed in the first step. I thought the authors could have fitted a deterministic model to the daily number of new infections / hospitalisations / ventilations / deaths by generating a likelihood function from these measures and running a Monte Carlo Markov Chain to estimate the parameters. What made the ABC approach more relevant?

From the Appendix tables 4, 5 and 6, the authors use very informative priors on most of their parameters (especially alpha, delta, and kappa). I think the authors should compare the prior to the posterior distribution for these parameters to highlight whether the fitting procedure was different from the prior assumptions.

The authors mention that Figure 2 “demonstrates that good model fits are achieved in all compartments across time.” I am not certain I find all the panels of Figure 2 convincing (for example the time series of “New Deaths” and “New in Hospital” show a lot of daily variations, which makes it harder to evaluate whether the fit is convincing). Could the authors aggregate the data and the simulation by week and show the match between the weekly time series? This could remove part of the dispersion observed in Figure 2 and make it easier for the reader to compare the inferred time series to the data.

In the Appendix (subsection 2.3.1), the authors explain the summary statistics used for their ABC approach. I am not sure I understood the last notation. Did they use the total number of cases infected, hospitalised, ventilated and deceased (before 15th-25th March), along with the number of cases that recovered before 4th April? If so, why did they only include the number of recovered cases early in the outbreak?

3/ Some clarifications on the risk model and the uncertainty of the estimates are needed

I am not familiar with the JAM method the authors used to compute the conditional risk effects. I believe they should add a couple of sentences to explain how this method matches the two inputs it uses.

In line 316, the authors state that “The independent effect of comorbidities and obesity attenuate with increasing severity of disease, while that of age and smoking increase”. I believe the authors’ conclusions should reflect that the 95% CIs are quite large (especially for H|I), which makes this comparison seem excessive (eg: the CI of the condition RR of smoking on H|I is between .21 and 14.52).

The authors consider that BMI and age are ordinal variables, did the authors explore the idea of using different RR for each age category (ie the RR between 21-40 and 0-19 would be different from the RR between 41-59 and 21-40)? Would it be possible (and worth testing) that the risks of severe illness abruptly increase for the highest age group / BMI?

Finally, I thought the tables 2 and 3 were very hard to read and interpret. I do not really see what conclusions to draw from these. I think the authors should consider using a graphical representation rather than a table, or greatly reduce the number of rows / columns. Furthermore, the authors only show the median estimates, whereas they reported very large confidence intervals for some of the conditional RRs. I think they should report and reflect on the confidence intervals of these estimates.

4/ The scenario implemented could be more realistic

The authors currently consider 9 scenarios representing a combination of isolating a fraction of the individuals older than 65 years old (0%, 50% or 100%), and adopting different levels of NPIs (None, Moderate or Observed). I believe this idea is relevant, especially in the context of vaccination campaigns aimed at certain age groups, but most of the scenarios implemented here are unrealistic. Indeed, I think complete isolation of older people is improbable (multi-generational households, care homes..), and imagining a situation where uncontrolled transmission would trigger absolutely no change in behaviour (or policy) is also unthinkable. I believe there would be great value in a more consistent exploration of the impact of a gradually increasing proportion of older individuals being protected (or isolated), mixed with different moderate values of stringency of control measures.

In line 44 the authors state, “Results highlight […] the efficacy of targeted subpopulation-level policy interventions in LAC.” I do not think this sentence is in agreement with the results shown by the authors. Indeed, there was only one set of simulations where a lockdown was not implemented and the number of deaths was similar to the data, and it came at the cost of overwhelming hospitals. Furthermore, this result relied on a complete isolation of those 65+, which is unrealistic. I would argue the results highlight the efficacy of a complete lockdown in limiting transmission, which is also what the authors write in the abstract and in the discussion. Therefore, I think they should remove this sentence.

Minor points

Footnote P5 “The susceptible population does not decrease sizeably during the time period considered in this study.” According to the first panel of Figure 2, Up to 25% of the population was infected between March and November, do the authors think this could potentially impact the effective reproduction number? If not, what decrease would they consider to be sizeable?

L288-290: The authors mention the hospital and ICU capacity limits, I think it may be relevant to add these thresholds to the right panel of Figure 3, in order to facilitate the comparison between the simulated number of hospitalisations and the maximum capacity.

L345-351: I think the explanation of how the risk profiles were grouped should come before Table 2 is described, since this is one of the columns of Table 2. I would therefore suggest moving these sentences up.

In the compartmental model, the authors estimate the parameter Pv, representing the proportion of hospitalised cases who need ventilation. I could not find the prior distribution or the estimated values of Pv in the Main Text, or in the supplement.

I did not find any plot of the values of r(t) estimated by the model, I think the authors should plot the distribution of all the parameters estimated.

In the Appendix, Subsection 2.3.1, does “D” stand for the Data or the death time series, I think the letter applies to both here, is it a mistake?

6. PLOS authors have the option to publish the peer review history of their article (what does this mean?). If published, this will include your full peer review and any attached files.

Reviewer #1: No

Reviewer #2: No

---

## [Author Response · Author response to Decision Letter 0]

12 May 2021

(Please see Response to Reviewers file in pdf)

Dear reviewers,

Thank you for the extremely careful review of our manuscript and providing such comprehensive and constructive reviews. We greatly appreciate the time and thoroughness that was devoted to its review and which has provided us with these comprehensive suggestions for improvement.

We have attempted to address every suggestion made by the two reviewers in the main text and the Appendix. We have documented the changes in the following point-by-point letter that responds to each comment raised by the two reviewers.

We would like to note that we followed Reviewer 2’s suggestion to split the original manuscript into two papers. The Part I paper, this paper, focuses on estimating the distribution of risk profiles among infections, hospital and ICU admission, and deaths, as well as the CFR and IFR associated with each profile. The Part II paper, which will be submitted at a later date, will focus on policy analyses that utilize the risk estimates from this (Part I) paper as parameter inputs. Splitting our manuscript into two papers allows us to focus each paper on a single objective, as a result making each paper easier for readers to follow. In the case of this Part I paper, focusing on the risk-profile-stratified estimates has allowed us to respond to many of the comments from both the reviewers and more clearly outline and describe our multi-step analysis plan in the methods section, including both the epidemic model and risk model.

Given that we will not be changing the methods or models underlying our analysis, but rather splitting the analysis and text into two papers, we still plan to submit our Part II paper for revision as part of the review process for the original manuscript (PONE-D-21-00882). Because of the great time and attention that both reviewers have already invested in understanding our methods, we hope that both manuscripts may be re-reviewed by the two reviewers. Please advise on how we should handle the re-submission or submission of Part II of the paper.

In the following, we begin by summarizing the major changes made in our revised manuscript. We then provide the point-by-point response letter. We note that we have repeated some of the content from the summary of major changes in the point-by-point responses, and apologize for the resulting length of this review letter; we thought it would be most clear for the editor and reviewers to follow the changes made to the manuscript by first reading a summary of all changes, and second reading the more detailed responses to each point. 

We look forward to your review and thank you in advance for your time and attention to our revised manuscript.

Kind regards,

Abigail Horn, Ph.D

David Conti, Ph.D

Summary of major changes (beyond splitting the paper into Part I and Part II)

Epidemic model (methods and results)

With more data available since the review, we have expanded the time frame of analysis to include the full first year of the epidemic in Los Angeles County (LAC), from March 1, 2020 through March 1, 2021, which spans 3 epidemic waves.

We changed data sources to a data source from the Los Angeles Times that provides data on cumulative infections and deaths, and the current number of COVID-19 patients in-hospital and in-ICU. Now, with direct observations of numbers in-ICU, we no longer need to include the ventilation compartment in the model and we can drop the parameter for the probability of ventilation (pV).

We have provided a more detailed explanation of our parameter estimation framework, which involved a two-step process of first using broad grid search and second approximate Bayesian computation (ABC) sampling. 

This approach was necessary because multiple parameter solutions to fit the model exist, meaning that estimated posterior distributions will be multimodal if allowed to vary over a wide prior parameter space. The two-step process was employed to define unimodal posterior distributions and achieve convergence in parameter estimates. The broad grid search step was used to identify possible regions for each parameter, from which we decided on a single mode. External data sources were used to specify the parameter range for the grid search. The ABC sampling step was used to estimate the final posterior distribution for each parameter, with a prior distribution informed by the chosen mode from the grid search step. 

We have added comments on the convergence of the ABC sampling step for the model parameters in the main text. We have also provided plots of the prior and posterior density distributions used in the ABC step and commented on their similarity / difference in the Appendix. The distribution ranges for the broad grid search step are referenced in the main text, with details and full specification provided in the Appendix.

Risk model (methods and results)

Based on the reviewers suggestion for more clarity, we have restructured our explanation of the risk model methods in the main text by breaking the explication into 6 steps, explaining each of the 6 steps separately, and adding a flow diagram illustrating each step (model/analysis) and its inputs (as data or a previous modeling step) and outputs. Supporting details and the mathematical specification of each step have been provided in the Appendix. As part of this revision, in Step 2 we provide a paragraph describing how the JAM method combines the correlation structure and marginal effects to get conditional effects, with further mathematical detail provided in the Appendix. 

We have recategorized the age factor, and now model the categories 0-18, 19-49, 50-64, 65-79, and 80+. These categories are now possible to model due to the availability of observed data on these categories using the LA Times data source; in the COVID-19 infections-by-age data source used in the original version of the manuscript, data on these more fine-grained categories was not available and we were not able distinguish between ages 60-79 from 80+.

We have added a step to the risk modeling approach whereby we estimate the conditional relative risk for age (conditional on the other three risk factors), for each of the three models of disease progression (H|I), (Q|H), (D|Q), by calibrating the risk model to observed COVID-19 data for LAC on deaths by age group. Now, our estimates of the frequency of each age group over infections and over deaths are both matched to observed data. This means we are no longer using JAM to estimate the conditional RR for age groups. Please see Step 5 of the risk model for details.

To visualize our estimates of the risk-profile-stratified probabilities of disease progression and CFR and IFR, we have removed Tables 2 and 3 which both reviewers commented were difficult to read and/or interpret. The purpose of Tables 2 and 3 was to convey the range of values that can be taken on by the risk-profile-stratified probabilities of disease progression and CFR / IFR. In the revision, we now convey this information graphically and tabularly by:

Including in the main text new figures that show the range of values that can be taken on by profiles falling within each age group; specifically, the mean, minimum, and maximum (as an error bar) of the probabilities/CFR/IFR across the composing risk profiles within each of the 5 age groups. 

Including in the main text a figure of the estimated median and the 95% CI of the CFR and IFR for the most populous (in the overall LAC population) risk profile within each age group. 

Including in the Appendix, as standard numerical tables, the median and the 95% CI of the risk-profile-stratified probabilities of disease progression in Appendix Tables 8-10, and the median and the 95% CI of the risk-profile-stratified case fatality rates and infection fatality rates in Appendix Tables 11-12, all across dates every two weeks from May 15 2020 - March 1 2021.

Reviewer #1: 

The manuscript was a pleasure to read, and the supplement was also well organized and helped me understand some of the statements made in the manuscript. Unfortunately I have not stayed up to date on the clinical epidemiology or policy issues surrounding COVID-19 so I will only be able to make comments on technical points.

Thank you for your careful read of our manuscript and very thorough and helpful comments in review. We are very glad to hear the manuscript was a pleasure to read, and that the supplement was helpful. We have attempted to address all of your thoughtful comments and questions in the main text and Appendix, and in the following provide a point-by-point explanation of all the changes that were made in response to your comments. Implementing your comments has greatly contributed to the readability and clarity of the methods used and conclusions drawn from the paper.

Substantive comments:

R1.1a. Line 72: Can the authors please describe why there is no A->D transition? I imagine COVID diagnosis obtained only through autopsy is not common, but some explanation here would help.

Thank you for asking for clarification on this point; we agree that although COVID-19 diagnosis through autopsy is rare, and explanation justifying our assumption is warranted.

In our model, the only route to death is through an observed infection, followed by hospitalization and ICU care, meaning we do not model individuals that die from COVID-19 illness at home rather than at a point-of-care. We justify this assumption because the majority of confirmed COVID-19 deaths cases result from individuals who die in SNF, hospital, or following a stay in hospital; this evidence comes from personal communication with a colleague, Professor Erika Garcia in the Department of Preventive Medicine at USC, who has analyzed COVID-19 mortality data for the state of California (see Garcia et al. 2021, Annals of Epidemiology, https://doi.org/10.1016/j.annepidem.2021.03.006). Although not included in that paper, Professor Garcia has analyzed the mortality data for CA to provide us with the estimate that 4%-9% of official COVID-19 deaths have occurred at home, across the three epidemic waves. 

Furthermore, we do not model a route to death for individuals without a confirmed COVID-19 infection, since record of confirmed COVID-19 infection (or probable based on clinical evidence) is needed to be classified as COVID-19 mortality. 

We have added this statement to the text under Section 2.1, beginning approximately line 125.

R1.1b. It may be worth stating upfront that the binary classification of infections as A or I only represents detection/entering the health system, and not asymptomatic/symptomatic, as in other modeling studies.

Thank you for this suggestion, we had provided this explanation in the Appendix but agree that it is an important distinction and needs to be included in the main text. We have included the following sentences to clarify this point, ~ line 78: 

We also include a compartment representing infectious individuals with unobserved and/or unconfirmed infections (A). I represents cases of infection that have tested positive for the SARS-CoV2 virus and are confirmed in the official register of infection case data. A represents cases that are symptomatic but do not appear in the confirmed case data, whether because they are asymptomatic, are symptomatic and do not get tested, or get tested and have a false negative result. 

R1.2. Line 77: I would appreciate some more explanation of what "proxy with error" means, with respect to the Q->V transition. In particular, what real process when a patient enters the ICU does that compartment represent? Does it mean that someone is only recorded as an ICU COVID patient when they are given a ventilator?

In the revision, we have removed the ventilation compartment from the model. We had originally included the ventilation compartment because we had available data on the numbers of patients on ventilation but not in the ICU, whereas our risk model probabilities were based on the progression from general hospital admittance to being advanced to critical care (in ICU). As noted in the Summary at the beginning of this letter, we have since obtained data on the number of COVID-19 patients in Los Angeles County in the ICU, and so removed the ventilation compartment.

R1.3. Section 2.1.1: if beta(t) is the parameter being estimated by ABC, what is the relationship of that estimated curve to mu(t) in the previous paragraph (starting on line 90)?

Thank you for asking for clarification on the relationship between beta(t) and mu(t). This comment and comment R1.7 below helped us to see that this relationship was not made sufficiently clear in the main text. To address this, in the revision we have removed the parameter mu(t) and consider only the time-varying infection rate, beta(t), and time-varying reproductive number, R(t). Now, when interventions are put in place that change the transmission rate, instead of estimating modifications to mu(t) to represent modifications in beta(t) and R(t), we discuss modifications to the time-varying parameters beta(t) and R(t) directly. 

R1.4. Line 124: I would appreciate some comments from the authors on convergence (or not) of the ABC sampling for the 6 parameters in the main text. If time permits, perhaps trace plots or histograms in the Appendix would be a nice addition.

We have added comments on the convergence of the ABC sampling for the model parameters in the main text (Section 3.1) and also provided density plots of the final distributions in the Appendix (Section 3.3). 

This was a very helpful suggestion, as it helped us to see that the parameter estimation framework was not fully explicated in the paper. In the revision we have provided more context on the parameter estimation process, as follows below (in Methods Section 2.1.1): 

Due to the relationships between parameters in the model formulation, multiple parameter solutions to fit the model exist. This means that estimated posterior distributions will be multimodal if allowed to vary over a wide prior parameter space. We use a two-step process to define unimodal posterior distributions and achieve convergence in parameter estimates by using broad grid search followed by approximate Bayesian computation (ABC) sampling. We first perform the broad grid search to identify possible regions for each parameter, from which we decide on a single mode. External data sources were used to specify the parameter range for the grid search (Appendix Section 2.1). Second, we use ABC sampling to estimate the final posterior distribution for each parameter with a prior distribution informed by the chosen mode from the grid search step. Specifically, we define the ABC prior as a normal distribution with 95% of its values lying within 25% of the mean value of the chosen mode; for example, if the mean of a chosen mode for parameter X is determined to be 0.1, then the prior distribution for X will be a normal distribution with standard deviation of 0.01, chosen such that Pr(0.075 < X < 0.125) ≈ 95%. 

Then, in the main text Results Section 3.1, we added the following comments on convergence of the ABC sampling for the model parameters, and provided density plots of the final distributions in the Appendix (Section 3.3): 

The two-step parameter estimation approach (broad grid search to select a single mode of each parameter, followed by approximate Bayesian computation (ABC) using a prior distribution specified around that single mode) achieved convergence in posterior densities. Convergence is not reached for the broad grid search step, with multi-modal distributions returned for each parameter (not shown). By specifying a narrow prior distribution around a mode chosen from the broad grid search sampling, convergence around a dominant single mode is achieved in the final posterior density returned by the ABC sampling step (see Appendix Section 3.3 for density plots of prior and posterior distributions).

R1.5. Line 161: For those of us unfamiliar with JAM, a sentence describing the assumptions needed to combine the correlation structure and marginal effects to get conditional effects would be a valuable addition.

One major change in our revision was to restructure our explanation of the risk model methods in the main text by breaking it into 6 steps, explaining each of the 6 steps separately, and adding a flow diagram illustrating each step (model/analysis) and its inputs (as data or a previous modeling step) and outputs. Supporting details and the mathematical specification of each step have been provided in the Appendix. As part of this revision, in Step 2 we provide a couple of sentences describing how JAM combines the correlation structure and marginal effects to get conditional effects; please see the main text Section 2.2.1 ~line 300, and the Appendix Section 5.3.

R1.6. Line 187: I understand that this accounts for uncertainty in estimated parameters and stochastic variation in trajectories, but does it also account for uncertainty in the estimates from the risk model?

This is a good question and the answer is no, it does not. We estimate the risk-profile-stratified probabilities of disease progression as fixed values (that are mean-centered on the means of the population-wide alpha, kappa, and delta), so the uncertainty in the risk-profile-stratified CFR and IFR comes only from the stochasticity in the epidemic model estimates of the population-wide infection and death timeseries. We have added the following line to the text to make this clear to the reader:

This process [of estimating the risk-profile-stratified CFR and IFR] therefore accounts for the uncertainty in the estimated parameters and stochasticity in the epidemic model, but not from the risk model estimates.

R1.7. Line 192: How does changing R(t) adjust beta(t) and mu(t)? I'm still somewhat unclear on the relationship of these quantities.

As noted in response to your comment R1.3 above, this comment helped us to see that the relationship between R(t), beta(t), and mu(t) was not made sufficiently clear in the main text, and so in the revision we have removed the parameter mu(t) and instead deal only with the time-varying infection rate, beta(t), and time-varying reproductive number, R(t). Now, when interventions are put in place that change the transmission rate, instead of estimating modifications to mu(t) to represent modifications in beta(t) and R(t), we discuss modifications to the time-varying parameters beta(t) and R(t) directly. 

Minor comments:

R1m.1. Abstract: CFR is used before it is defined

Thank you for pointing this out, we have corrected it.

R1m.2. Line 70: please remind readers which compartments S,E,R are.

We have added much detail to our explanation of the epidemic model in the main text, which we believe has sufficiently clarified the meaning of each compartment, including S, E, and R; please see the main text section 2.1.

R1m.3. Line 105: there is an inconsistent use of subscript t and function of t for parameters alpha, kappa, delta between text and Figure 1. 

We have corrected this in the revision and now use subscript t for all parameters, while saving function of t for all state variables.

Also, p_v is not defined anywhere, what is it?

As noted above, we have removed the ventilation compartment from the model and the associated probability of ventilation p_v. We apologize for the lack of clarity in the original version of the manuscript.

R1m.4. Line 204: text references Fig 4a but looks like it should say Fig 4b.

In the revision, we have updated all figures, their captions, and in-text references. 

R1m.5. Line 228: text references Fig 4b but looks like it should say Fig 4a.

Wee have updated all figures, their captions, and in-text references. 

R1m.6. Tables 2 and 3 are very hard to read (too small), could they be enlarged somewhat?

We agree Tables 2 and 3 were hard to read and interpret. Reviewer 2 suggested that a graphical representation could be used to convey the information in Tables 2 and 3. In our revision, we have replaced Tables 2 and 3 with both graphical representations and standard numerical tables as follows.

The purpose of Tables 2 and 3 was to convey the range of values that can be taken on by the risk-profile-stratified probabilities of disease progression and CFR / IFR. To convey this information graphically, in the main text we have included Figures 5a-c and 6a-d. We show in Figures 5a - 5c the range of values that can be taken on by profiles falling within each age group. Specifically, these figures show the mean, minimum, and maximum (as an error bar) of the probabilities across the composing risk profiles within each of the 5 age groups, under each of the three risk models. Figures 6a - b show the mean, minimum, and maximum of the median CFR and IFR values that can be taken on by the risk profiles within each age group. We also plot the estimated median and the 95% CI for the most populous (in the overall LAC population) risk profile within each age group in Figures 6c - d.

We have also included, as standard numerical tables, the median and the 95% CI of the risk-profile-stratified probabilities of disease progression in Appendix Tables 8-10, and the median and the 95% CI of the risk-profile-stratified case fatality rates and infection fatality rates in Appendix Tables 11-12, all across dates every two weeks from May 15 2020 - March 1 2021.

Reviewer #2: 

In this study, the authors present an analysis of the COVID-19 transmission dynamics in Los Angeles county (LAC). They aim to estimate the probability of severe illness depending on the risk profile of the individuals, and analyse the impact different control measures could have had on the number of infected individuals. In order to carry out this analysis, the authors developed the following workflow:

1 - Estimate the population-wide proportion of hospitalised cases, proportion of hospitalisations leading to Intensive Care Unit (ICU) admission, proportion of deaths given ICU admission, proportion of reported cases, and reproduction number over time using a compartmental model and Approximate Bayesian Computation.

2 - Estimate the conditional relative risk of different factors on the proportion of hospitalisations, ICU admissions and deaths using marginal relative risks from the previous literature and Joint Analysis of Marginal summary statistics (JAM).

3 - Estimate the proportion of cases from different risk profiles using the marginal risk factor in LAC, and the age distribution the reported cases.

4 - Use the population-wide estimates from step 1, the conditional risk estimates from step 2 and the distribution of the risks profiles of the infected population from step 3 to deduce the risks of hospitalisation given infection, ICU admission given hospitalisation, and death given ICU admission for each risk profile.

5 - Generate simulations of outbreaks using different scenarios of Non-Pharmaceutical Interventions.

This paper mixes different complex methods and uses different publicly available data sources. I appreciate the time and effort the authors have spent to provide the code and make their analysis reproducible on a Github repository. I also want to highlight the thorough Appendix detailing the approach used in every step of the analysis.

Overall, I believe this is an interesting piece, which can provide important contributions to the field. Nevertheless, I think there needs to be some clarification on some stages of the study. It took me time to understand the justifications behind each step of the workflow, notably what outputs were needed for the final results, and the uncertainty of some of the estimated parameters.

Thank you for your complimentary words, and moreover for the great time and attention put into your extremely thorough and constructive review of our manuscript, resulting from your very careful read of both the main text and the Appendix. We have addressed all of your suggestions, which we believe have contributed in a major way to improving the quality of our article.

Before getting to major and minor points, I had an overarching comment on the paper:

R2.0/ Because of the number of different stages and models developed in the study, I think the message / workflow sometimes gets lost, or at least I got confused a number of times. I wonder whether the authors would consider splitting this analysis into two papers: One focused on estimating the distribution of risk profiles in infected cases and the probabilities of hospitalisation (ICU and death) associated with each profile (ie steps 2 to 4, and the first objective of this paper), and one focused on the simulations of different scenarios of NPIs and vaccine coverage using the risks profiles (mostly Step 5). This way, the different scenarios of NPI included in the simulations could be deepened and more realistic, and the authors could give more information in the Main Paper on the JAM method and logistic regression they implemented (and use the prior distribution on alpha, kappa and delta for their estimations inn Step 4). I do not think this is a requirement for publication, but I believe this would make it easier for the reader to follow the arguments the authors are presenting.

After carefully considering this suggestion, we decided to split the original manuscript into two papers, exactly as you suggested: The first paper, which is presented in this submitted revision, focuses on estimating the distribution of risk profiles among infected cases, hospitalized and ICU admitted patients, and deaths, as well as the CFR and IFR associated with each profile. The second paper, which we are still working on, will focus on policy analyses that utilize the risk estimates from the first paper (this paper) as parameter inputs. 

We agree that splitting our manuscript into two papers will allow us to focus each paper on a single objective, as a result making each paper easier for the reader to follow. In the case of this paper, focusing on the risk-profile-stratified estimates has allowed us to more clearly outline and describe our multi-step analysis plan in the methods section, as well as the findings from both the epidemic model and risk model.

Major points

R2.1/ Summarise the overall workflow in a Figure.

R2.1a/ In the summary of my review, I tried to summarise the workflow the authors implemented from the Main text and Appendix. Although I hope I understood it correctly, it took me time and a few read-throughs to figure out how and why the authors went from one stage to another. I think the authors should add a figure summarising each stage of the analysis, along with their input and output, and how they connect to one another. I believe this would be very helpful for the readers, and would prevent a lot of confusion.

You have provided an excellent summary of our workflow. Thank you for your very careful read and attention to the details of our paper that was required to understand all steps of our method. We have taken this point to heart and substantively edited the methods section describing the risk model in the main text (Section 2.2) and the Appendix (Appendix Part II) to outline, describe, and visualize in Figure 2 a workflow of all steps taken in our methodology.

R2.1b/ Along with that, the authors use a lot of different notations in each section of the Appendix. I believe they should summarise all the notations of Section 2 in a table (similar to what they did in Table 1, 2, and 3 of the Appendix). This would facilitate the reading and general understanding of their analysis.

Thank you for this helpful suggestion. We have included in Table 8 of the Appendix a table summarizing all notations used in developing the risk model, i.e. Part II of the Appendix. In the table we have distinguished between which notations represent definitions, observed data, and model-produced estimates. We agree with the reviewer that this will facilitate the reader’s understanding of the model.

R2.2/ The implementation of the compartmental model should be clearer

R2.2a/ I am not sure I understand why Approximate Bayesian Computation (ABC) was needed in the first step. I thought the authors could have fitted a deterministic model to the daily number of new infections / hospitalisations / ventilations / deaths by generating a likelihood function from these measures and running a Monte Carlo Markov Chain to estimate the parameters. What made the ABC approach more relevant?

We respond to this comment in two parts. First, we motivate our choice of employing a stochastic model instead of a deterministic model. We felt a stochastic model was needed to represent the stochasticity in the infection process especially when numbers are small in certain compartments. LAC is a large county made up of many composing cities and communities, each with their own epidemic processes unfolding at different rates with different peaks. The unified trend across these many cities/communities is one of substantive day-to-day variation in counts of new infections and deaths, since it is the composite of multiple different trends. Furthermore, it has been well established that there is stochasticity in the observation and reporting of infections, and to a lesser extent hospitalizations and deaths. Since we do not attempt to model these observation/reporting processes themselves, our model is not capable of mechanistically identifying the “true” trend of each state variable from the observed data. We therefore believe a stochastic model is merited, which is able to represent a range of scenarios and confidence intervals by simulating it across stochastic runs. 

Second, we motivate our choice of ABC sampling to estimate parameters of our stochastic model. First and foremost, a tractable likelihood function was not possible for our model, which involves five free and interacting parameters (or six, if the starting time is included). Therefore, a likelihood-free method of parameter estimation was required. ABC provides a number of benefits that make it particularly well suited to this application, not only allowing estimation of model parameters when a likelihood function is intractable but also providing a suitable framework when: prior information and/or assumptions are available about the distribution or range of values each parameter may take; multiple data features are available for model estimation, e.g. infection, hospitalization, and death counts; and data is missing, partially observed, or uncertain, such as unreliable early infection surveillance data. 

Finally, we note that while ABC is a computation-heavy approach, simulating our stochastic compartmental model is computationally cheap, and so we are able to employ ABC without any computational issues in this setting.

R2.2b/ From the Appendix tables 4, 5 and 6, the authors use very informative priors on most of their parameters (especially alpha, delta, and kappa). I think the authors should compare the prior to the posterior distribution for these parameters to highlight whether the fitting procedure was different from the prior assumptions.

This is a fair point, we agree that comments on the similarity between the prior and posterior distributions from ABC parameter estimation should be provided. First, it is important to note (as we have now noted in the text) that our parameter estimation process involved a first step of a broad grid search of each parameter over a wide range. This step returned the modes around which we defined prior distributions for the ABC sampling step. Final posterior distributions for each parameter were quite distinguished from the initial range of values sampled in the broad grid search step.

With that said, we felt it would still be instructive to compare the plots of prior distributions used in the ABC sampling step to the final posterior distributions produced for each parameter. In the Appendix Section 3.3, we have provided the density plots for the prior and posterior distributions from ABC parameter estimation. Comparing the distributions, it can be seen that although the prior distributions used in the ABC step are narrow (with 95% of a prior parameter's value lying within 25% of the mean of the chosen mode from broad grid search), they are not too narrow to allow the posterior distributions to take a different shape such that all posterior distributions differ slightly from the prior. The mean of the posterior is not exactly aligned with the mean of the prior, and the standard deviation becomes narrower. We have provided these observations in the text in Appendix Section 3.3.

R2.2c/ The authors mention that Figure 2 “demonstrates that good model fits are achieved in all compartments across time.” I am not certain I find all the panels of Figure 2 convincing (for example the time series of “New Deaths” and “New in Hospital” show a lot of daily variations, which makes it harder to evaluate whether the fit is convincing). Could the authors aggregate the data and the simulation by week and show the match between the weekly time series? This could remove part of the dispersion observed in Figure 2 and make it easier for the reader to compare the inferred time series to the data.

First, as noted in the Summary, we have added a new data source that provides current numbers in-hospital and in-ICU; trends for these parameters are undispersed and clearly interpretable for the reader. 

Regarding aggregating infections and deaths in presenting plots for the reader, this is the one point on which we do not agree with the reviewer. We believe that it is necessary to present the true stochasticity in the data in our figures, as this is the stochastic data we were fitting our model to. As noted in our response to comment R2.2a, this stochasticity in the observed and reported data is the closest to the “truth” of the infection process we have, without attempting to estimate the error in the observation and reporting processes, which is not a feature of our modeling framework. Additionally, the stochasticity in the data is also representative of the fact that the epidemic trend in LAC is in fact a composite of many epidemic trends occurring in the cities/communities that make up the county. For these reasons, we felt it necessary to use a stochastic model, to fit the stochastic model to the stochastic data, and to present the stochastic data to the reader in the plots. In sum, an aggregate statistic created on the data could add bias or error to the data being fit to, and so we choose to use the raw data, which the stochastic model is capable of estimating to. 

This all said, we have produced for this review figures that shows model estimates of new infections and new deaths plotted against the data as a 7-day running average (see below). However for the reasons given above, we would prefer to provide the reader with the view of the raw observed data used to fit the model.

R2.2d/ In the Appendix (subsection 2.3.1), the authors explain the summary statistics used for their ABC approach. I am not sure I understood the last notation. Did they use the total number of cases infected, hospitalised, ventilated and deceased (before 15th-25th March), along with the number of cases that recovered before 4th April? If so, why did they only include the number of recovered cases early in the outbreak?

Thank you for this question, which identified a mistake in the Appendix in Section 2.3.1 that was left over from an earlier draft of the manuscript. Recovered cases are not used in estimating the model and should not be included in the summary statistic function; we have removed this mention. 

R2.3/ Some clarifications on the risk model and the uncertainty of the estimates are needed

R2.3a/ I am not familiar with the JAM method the authors used to compute the conditional risk effects. I believe they should add a couple of sentences to explain how this method matches the two inputs it uses.

Step 2 out of our 6-step restructuring of the methods section describing the risk model deals with the implementation of JAM to estimate the conditional relative risks. In the description of Step 2 in both the main text and the Appendix, we have provided a paragraph describing how JAM combines the correlation structure and marginal effects to get conditional effects; please see the main text Section 2.2.1 ~line 300, and the Appendix Section 5.3 for additional mathematical formalization.

R2.3b/ In line 316, the authors state that “The independent effect of comorbidities and obesity attenuate with increasing severity of disease, while that of age and smoking increase”. I believe the authors’ conclusions should reflect that the 95% CIs are quite large (especially for H|I), which makes this comparison seem excessive (eg: the CI of the condition RR of smoking on H|I is between .21 and 14.52).

This is a fair point, we have tempered our statement by replacing the text quoted above with the following:

We observe that the independent effect of comorbidities and obesity attenuate with increasing severity of disease; smoking may increase with age, however a very wide confidence interval for (H|I) makes this conclusion tentative. 

R2.3c/ The authors consider that BMI and age are ordinal variables, did the authors explore the idea of using different RR for each age category (ie the RR between 21-40 and 0-19 would be different from the RR between 41-59 and 21-40)? Would it be possible (and worth testing) that the risks of severe illness abruptly increase for the highest age group / BMI?

This is a great question and in fact we now do estimate a different conditional RR for each age category. This was achieved by adding a step to the risk modeling approach (Step 5) whereby we estimate the conditional relative risk for age for each of the three models of disease progression (H|I), (Q|H), (D|Q) separately from estimating the conditional RR for BMI, smoking, and any comorbidity. Specifically, we estimate the conditional RR for age by calibrating the risk model to observed COVID-19 data for LAC on deaths by age group. Now, our estimates of the frequency of each age group over infections and over deaths are both matched to observed data. In this way, we are able to estimate a different RR for each category.

We have also added a 5th age category to our age category set, and now model the categories 0-18, 19-49, 50-64, 65-79, and 80+. This enables us to model the sharp increase in risk for individuals in the higher age groups. 

R2.3d/ Finally, I thought the tables 2 and 3 were very hard to read and interpret. I do not really see what conclusions to draw from these. I think the authors should consider using a graphical representation rather than a table, or greatly reduce the number of rows / columns. Furthermore, the authors only show the median estimates, whereas they reported very large confidence intervals for some of the conditional RRs. I think they should report and reflect on the confidence intervals of these estimates.

We agree that Tables 2 and 3 were hard to read and interpret and in our revision, we have replaced Tables 2 and 3 with both graphical representations and standard numerical tables as follows.

The purpose of Tables 2 and 3 was to convey the range of values that can be taken on by the risk-profile-stratified probabilities of disease progression and CFR / IFR. To convey this information graphically, in the main text we have included Figures 5a-c and 6a-d. We show in Figures 5a - 5c the range of values that can be taken on by profiles falling within each age group. Specifically, these figures show the mean, minimum, and maximum of the probabilities across the composing risk profiles within each of the 5 age groups, under each of the three risk models. Figures 6a - b show the mean, minimum, and maximum of the median CFR and IFR values that can be taken on by the risk profiles within each age group. 

We have also reported and commented on the confidence intervals of the median estimates. We plot the estimated median and the 95% CI (vs. the minimum and maximum profile by median only, as in 6a-b) for the most populous (in the overall LAC population) risk profile within each age group in Figures 6c - d. In the text describing Figures 6c-d, we have commented on the confidence intervals observed in the figures.

We have also included, as standard numerical tables, the median and the 95% CI of the risk-profile-stratified probabilities of disease progression in Appendix Tables 8-10, and the median and the 95% CI of the risk-profile-stratified case fatality rates and infection fatality rates in Appendix Tables 11-12, all across dates every two weeks from May 15 2020 - March 1 2021.

R2.4/ The scenario implemented could be more realistic

R2.4a/ The authors currently consider 9 scenarios representing a combination of isolating a fraction of the individuals older than 65 years old (0%, 50% or 100%), and adopting different levels of NPIs (None, Moderate or Observed). I believe this idea is relevant, especially in the context of vaccination campaigns aimed at certain age groups, but most of the scenarios implemented here are unrealistic. Indeed, I think complete isolation of older people is improbable (multi-generational households, care homes..), and imagining a situation where uncontrolled transmission would trigger absolutely no change in behaviour (or policy) is also unthinkable. I believe there would be great value in a more consistent exploration of the impact of a gradually increasing proportion of older individuals being protected (or isolated), mixed with different moderate values of stringency of control measures.

As discussed, we have removed the policy analysis from this Part I paper. This will be the focus on the Part II paper, and we will be sure to take these comments into consideration. We will be implementing more realistic scenarios in terms of both NPIs and protecting those at higher risk. 

Minor points

R2.4b/ In line 44 the authors state, “Results highlight […] the efficacy of targeted subpopulation-level policy interventions in LAC.” I do not think this sentence is in agreement with the results shown by the authors. Indeed, there was only one set of simulations where a lockdown was not implemented and the number of deaths was similar to the data, and it came at the cost of overwhelming hospitals. Furthermore, this result relied on a complete isolation of those 65+, which is unrealistic. I would argue the results highlight the efficacy of a complete lockdown in limiting transmission, which is also what the authors write in the abstract and in the discussion. Therefore, I think they should remove this sentence.

Thank you for pointing this out. Although this sentence and the discussion of the policy analysis is no longer included in this Part I paper, we agree with the reviewer on this point and will not be including the highlighted sentence / conclusion in the Part II paper. 

R2m.1/ Footnote P5 “The susceptible population does not decrease sizeably during the time period considered in this study.” According to the first panel of Figure 2, Up to 25% of the population was infected between March and November, do the authors think this could potentially impact the effective reproduction number? If not, what decrease would they consider to be sizeable?

In the revision, we have plotted and reflected on both the R(t) based on behavior alone and the effective R(t) (R_eff(t)) that takes into account the diminishing size of the susceptible population. The comparison of R(t) and R_eff(t) allows us to say that an appreciable divergence between the two began as early as the beginning of July, following the first half of the second wave. This was an important observation, since R_eff(t) had dropped back below 1 by mid-July, two weeks earlier than R(t) did.

Furthermore, as the cumulative infected population grew with the major surge of the third epidemic wave, we estimate this fraction grew to 40-60% by March 1, 2021, which would be sizeable by anyone’s definition. 

R2m.2/ L288-290: The authors mention the hospital and ICU capacity limits, I think it may be relevant to add these thresholds to the right panel of Figure 3, in order to facilitate the comparison between the simulated number of hospitalisations and the maximum capacity.

Thank you for this suggestion. We have plotted the hospital and ICU capacity limits in the figure showing current cases in-hospital and in-ICU, which we have moved to the Appendix Figure 5.

R2m.3/ L345-351: I think the explanation of how the risk profiles were grouped should come before Table 2 is described, since this is one of the columns of Table 2. I would therefore suggest moving these sentences up.

We agree this would have been a helpful suggestion, however in our revision we have omitted the grouping of the risk profiles into the 5 risk categories; we thought this added little value to the overall interpretation of the calculated risk values.

R2m.4/ In the compartmental model, the authors estimate the parameter Pv, representing the proportion of hospitalised cases who need ventilation. I could not find the prior distribution or the estimated values of Pv in the Main Text, or in the supplement.

Wee have removed the ventilation compartment from the model and the associated probability Pv. We had originally included the ventilation compartment because we had available data on the numbers of patients on ventilation but not in the ICU, whereas our risk model probabilities were based on the progression from general hospital admittance to being advanced to critical care (in ICU). We have since obtained data on the number of COVID-19 patients in Los Angeles County in the ICU, and so removed the ventilation compartment.

R2m.5/ I did not find any plot of the values of r(t) estimated by the model, I think the authors should plot the distribution of all the parameters estimated.

We have included a plot of the model-estimated r(t) in the revision (Figure 4c).

R2m.6/ In the Appendix, Subsection 2.3.1, does “D” stand for the Data or the death time series, I think the letter applies to both here, is it a mistake?

Thank you for this question and for noticing this mistake. In the highlighted section, “D” was meant to indicate the data, but we agree that this notation cannot be used since “D” is reserved for the death timeseries. In the revision we have used the notation Φ to represent the data.

---

## [Decision Letter · Decision Letter 1]

8 Jun 2021

An integrated risk and epidemiological model to estimate risk-stratified COVID-19 outcomes for Los Angeles County: March 1, 2020 - March 1, 2021

PONE-D-21-00882R1

Dear Dr. Horn,

We’re pleased to inform you that your manuscript has been judged scientifically suitable for publication and will be formally accepted for publication once it meets all outstanding technical requirements.

Kind regards,

Martial L Ndeffo Mbah, Ph.D

Academic Editor

PLOS ONE

Additional Editor Comments (optional):

Reviewers' comments:

Reviewer's Responses to Questions

**Comments to the Author**

1. If the authors have adequately addressed your comments raised in a previous round of review and you feel that this manuscript is now acceptable for publication, you may indicate that here to bypass the “Comments to the Author” section, enter your conflict of interest statement in the “Confidential to Editor” section, and submit your "Accept" recommendation.

Reviewer #1: All comments have been addressed

Reviewer #3: All comments have been addressed

2. Is the manuscript technically sound, and do the data support the conclusions?

Reviewer #1: Yes

Reviewer #3: Yes

3. Has the statistical analysis been performed appropriately and rigorously? 

Reviewer #1: Yes

Reviewer #3: Yes

4. Have the authors made all data underlying the findings in their manuscript fully available?

Reviewer #1: Yes

Reviewer #3: Yes

5. Is the manuscript presented in an intelligible fashion and written in standard English?

Reviewer #1: Yes

Reviewer #3: Yes

6. Review Comments to the Author

Reviewer #1: (No Response)

Reviewer #3: All comments and criticisms raised in the previous review round have been addressed by authors, who should be commended for their efforts.

7. PLOS authors have the option to publish the peer review history of their article (what does this mean?). If published, this will include your full peer review and any attached files.

Reviewer #1: No

Reviewer #3: No

---

## [Editor Report · Acceptance letter]

15 Jun 2021

PONE-D-21-00882R1 

An integrated risk and epidemiological model to estimate risk-stratified COVID-19 outcomes for Los Angeles County: March 1, 2020 - March 1, 2021 

Dear Dr. Horn:

I'm pleased to inform you that your manuscript has been deemed suitable for publication in PLOS ONE. Congratulations! Your manuscript is now with our production department. 

Kind regards, 

on behalf of

Dr. Martial L Ndeffo Mbah 

Academic Editor

PLOS ONE